# IGF1R regulates retrograde axonal transport of signalling endosomes in motor neurons

Alexander D Fellows[1,†], Elena R Rhymes[1,†], Katherine L Gibbs[1], Linda Greensmith[1] & Giampietro Schiavo[1,2,3,*] iD

## Abstract

**Signalling endosomes are essential for trafficking of activated ligand–receptor complexes and their distal signalling, ultimately leading to neuronal survival. Although deficits in signalling endosome transport have been linked to neurodegeneration, our understanding of the mechanisms controlling this process remains incomplete. Here, we describe a new modulator of signalling endosome trafficking, the insulin-like growth factor 1 receptor (IGF1R). We show that IGF1R inhibition increases the velocity of signalling endosomes in motor neuron axons, both *in vitro* and *in vivo*. This effect is specific, since IGF1R inhibition does not alter the axonal transport of mitochondria or lysosomes. Our results suggest that this change in trafficking is linked to the dynein adaptor bicaudal D1 (BICD1), as IGF1R inhibition results in an increase in the *de novo* synthesis of BICD1 in the axon of motor neurons. Finally, we found that IGF1R inhibition can improve the deficits in signalling endosome transport observed in a mouse model of amyotrophic lateral sclerosis (ALS). Taken together, these findings suggest that IGF1R inhibition may be a new therapeutic target for ALS.**

**Keywords** axonal transport; cytoplasmic dynein; IGF1R; signalling endosome; tetanus toxin

**Subject Categories** Membranes & Trafficking; Neuroscience; Signal Transduction

## Introduction

Signalling endosomes are responsible for the trafficking and distal signalling of activated growth factor receptors in all cell types. In neurons, the long-range retrograde axonal transport of signalling endosomes containing neurotrophins is vital for synaptic plasticity, axon growth and nerve repair [1]. The spatial distribution and trafficking of these organelles, together with mitochondria, RNA granules and lysosomes, is vital for neuronal maintenance and survival.

Accordingly, perturbations in these processes are associated with neurodevelopmental and neurodegenerative diseases [2]. Deficits in the axonal transport of signalling endosomes have been detected in several animal models of neurodegenerative diseases, such as amyotrophic lateral sclerosis (ALS), Alzheimer's disease and Huntington's disease [3–6]. Since the molecular determinants of these deficits are currently unknown, it is important to elucidate the mechanisms regulating the trafficking of these organelles.

The retrograde, minus-end directed, transport of signalling endosomes in the axon is driven by cytoplasmic dynein. Intriguingly, by itself this molecular motor complex is mainly non-processive and therefore requires an array of adaptor proteins and cofactors to carry out its myriad of functions. This includes dynactin, which is necessary for the processive movement of dynein [7], and molecular adaptors, which confer the ability to bind specific cargo and can alter the kinetics of the dynein motor [8]. How these complexes assemble and are regulated in different cellular compartments, is still largely unclear.

Protein kinases have been shown to play an important role in axonal transport, altering trafficking dynamics by either directly influencing motor protein activity or regulating how they bind to the microtubule network [9–11]. Furthermore, abnormal activation of specific protein kinases has been shown to be causative for transport deficits detected in an animal model of ALS [12]. Therefore, protein kinases represent a promising avenue to explore the mechanisms regulating axonal transport, which in turn might lead to the identification of potential therapeutic targets for several human disorders, including neurodegenerative diseases.

In this study, we screened a library of kinase inhibitors and identified IGF1R as a regulator of the retrograde transport of signalling endosomes. IGF1R is a transmembrane tyrosine kinase receptor that is essential for neuronal development and survival. Furthermore, it has been found to play a role in neuronal polarization and is neuroprotective after injury [13–15]. Here, we found that genetic and pharmacological inhibition of IGF1R in motor neurons results in a significant increase in the retrograde velocity of signalling endosomes, whilst treatment of neurons with IGF1, an IGF1R agonist, slowed down axonal transport. Intriguingly, this was not due to a direct effect on the dynein motor as the retrograde transport of other cargoes was unaffected. Instead, we saw a significant increase in the

1  Department of Neuromuscular Diseases, UCL Queen Square Institute of Neurology, London, UK
2  UK Dementia Research Institute at UCL, London, UK
3  Discoveries Centre for Regenerative and Precision Medicine, University College London Campus, London, UK
   *Corresponding author. Tel: +44 7918 738393; E-mail: giampietro.schiavo@ucl.ac.uk
   †These authors contributed equally to this work

*de novo* synthesis of the dynein adaptor protein BICD1 in the axon, which may account for the change in velocity of retrograde signalling endosomes observed in this study.

# Results

### A kinase inhibitor screen reveals a novel modulator of retrograde axonal transport

To identify novel modulators of axonal transport, we tested a small-molecule kinase inhibitor library, using the accumulation of the axotoxic binding fragment of tetanus toxin ($H_cT$) and an antibody directed against the extracellular domain of the p75 neurotrophin receptor (α-p75$^{NTR}$) in the soma, as a biological readout of axonal transport [12]. This validated assay has been shown to be sufficiently sensitive to detect changes in retrograde axonal transport [12,16,17]. In this study, we used ES cell-derived motor neurons expressing green fluorescent protein (GFP) driven by the Hb9 homeobox gene enhancer, which allowed us to unequivocally identify motor neurons and overcome the intrinsic cellular heterogeneity found in primary ventral horn spinal cord cultures. Using a reliable, nonbiased automatic protocol [12], we screened a library of kinase inhibitors, with all compounds initially being tested at a concentration of 2 μM. Compounds that increased the mean signal intensity of $H_cT$ and α-p75$^{NTR}$ in the neuronal soma by at least three standard deviations above control levels (DMSO; Fig 1A, yellow rectangle) were classified as potential enhancers of retrograde axonal transport. Erythro-9-(2-hydroxy-3-nonyl) adenine (EHNA), an established inhibitor of cytoplasmic dynein, which blocks the retrograde transport of $H_cT$ along the axon [18], was used as a negative control. EHNA successfully decreased both $H_cT$ and α-p75$^{NTR}$

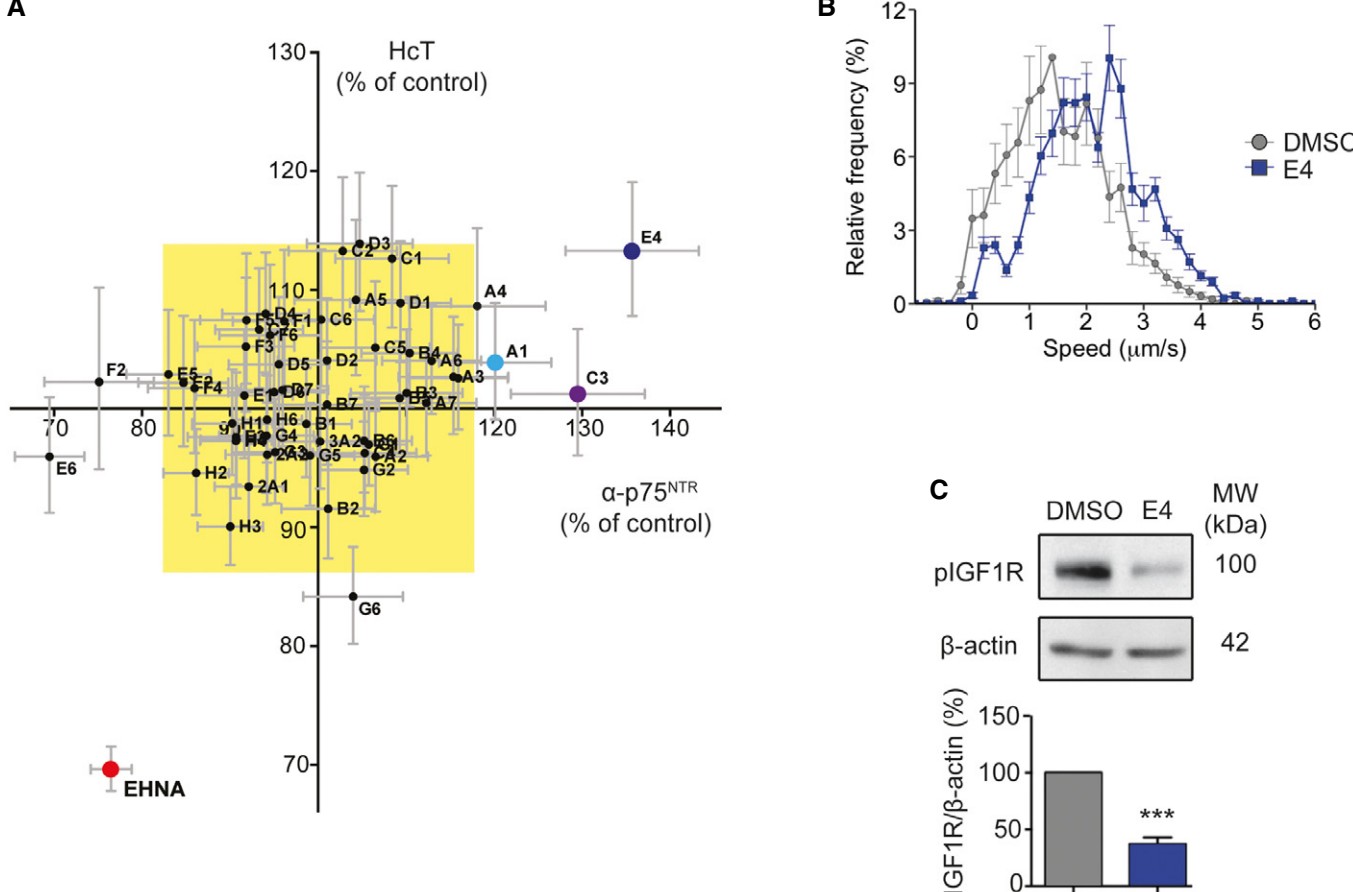

**Figure 1. Identification of E4 as a modulator of retrograde axonal transport.**

A  Small-molecule kinase inhibitor screen. Data are shown as an XY plot of normalized mean staining intensity of Alexa Fluor 555-$H_cT$ versus α-p75$^{NTR}$. The α-p75$^{NTR}$ was detected using an Alexa Fluor 647-conjugated donkey anti-rabbit secondary antibody [12]. Compounds that increased the accumulation of $H_cT$ or α-p75$^{NTR}$ by at least three standard deviations (yellow box) were classified as active compounds (A1—light blue; C3—purple; E4—dark blue). The negative control (EHNA) is shown in red (≥ 25 cell bodies were imaged per condition, N = 3 independent experiments).

B  Speed distribution profile of PMN treated with 0.5 μM E4 or DMSO for 30 min. Graph shows an increase in the rate of axonal transport upon E4 treatment (DMSO, 153 endosomes, 13 axons; E4, 165 endosomes, 13 axons; N = 4 independent experiments; data shown are mean ± SEM).

C  PMN treated with 0.5 μM E4 for 30 min showed a significant decrease in phospho-IGF1R levels compared to controls. Protein levels were normalized to β-actin (***P < 0.001, Student's t-test, N = 3 independent experiments; data shown are mean ± SEM).

accumulation, further validating our approach (Fig 1A). We identified three active compounds in our screen (Fig 1A; A1, C3 and E4), with E4 being the most effective at the concentration tested. Therefore, this compound was taken forward in this study; the effects of compounds A1 and C3 have been previously described [12]. Further information can be found, along with a complete list of the kinase inhibitor screen in Gibbs *et al* [12].

To assess whether E4 was a true enhancer of axonal retrograde transport, we also tested its effects in a live *in vitro* axonal transport assay performed in primary motor neurons (PMNs) using fluorescent $H_cT$ [19]. In PMN treated with 0.5 μM E4 at 6–7 days *in vitro* (DIV) for 30 min, we observed a substantial increase in the retrograde velocity of signalling endosomes (Fig 1B). Although E4 (GSK1713088A; CHEMBL517171) has been previously reported to inhibit IGF1R [20], we confirmed its effect in motor neurons by treating PMN cultures with 0.5 μM E4 and quantifying the levels of phosphorylated IGF1R (pIGF1R; Tyr1161/1165/1166) by immunoblotting. We found a significant decrease in pIGF1R under these conditions (Fig 1C). Taken together, these data indicate that E4 modulates the retrograde transport of signalling endosomes by inhibiting IGF1R, suggesting that this signalling pathway is involved directly or indirectly in the regulation of axonal transport.

**Pharmacological inhibition of IGF1R increases axonal signalling endosome motility *in vitro***

To confirm the role of IGF1R in retrograde axonal transport, we next tested whether the changes in signalling endosome velocity were due to any off-target effects of E4. To this end, we used an established IGF1R inhibitor, picropodophyllotoxin (PPP). PPP works by inhibiting the phosphorylation of Tyr1136 [21], which is crucial for receptor activation, and has very limited *in vitro* and *in vivo* toxicity [22,23]. We therefore measured the effect of PPP at 1 μM in a live *in vitro* retrograde axonal transport assay (Fig 2A and B, Appendix Fig S1A–D). PPP treatment caused a significant increase in the mean retrograde signalling endosome speed, with a velocity of 1.77 ± 0.06 μm/s compared to 1.55 ± 0.05 μm/s in control conditions (Fig 2C, Movie EV1). This increase was not caused by a decrease in pausing events (17 ± 7.2% versus 14.7 ± 6.5%, DMSO versus PPP, respectively; Fig 2D). Instead, this change was driven by an increase in instantaneous velocities, as shown in Fig 2G.

Since IGF1R inhibition results in an increase of the speed of retrograde axonal transport, it is possible that activation of IGF1R may lead to a decrease in the transport velocity of signalling endosomes. Accordingly, treatment of PMN with IGF1 (50 ng/ml) results in a decrease in the mean velocity of retrograde signalling endosomes (1.94 ± 0.06 μm/s to 1.54 ± 0.07 μm/s; Fig 2E, Movie EV2). This decrease was due to both an increase in pausing (11.8 ± 1.8% to 20.9 ± 2.9%; Fig 2F) and a decrease in instantaneous velocities (Fig 2H). Importantly, IGF1R was also found to be enriched at growth cones (Fig EV1A–C). As microtubule plus-ends, which are the initiation sites for retrograde transport, are located at growth cones and axonal endings, this result adds further evidence that this receptor may play a role in this process.

Upon IGF1 binding, IGF1R has been shown to activate two main downstream signalling pathways, PI3K/Akt and MAPK/Erk [24]. To elucidate whether IGF1R-mediated modulation of the retrograde trafficking of signalling endosomes is due to activation of these

signalling cascades, we measured the activation of key components of these pathways following incubation of PMN with 1 mM PPP or 50 ng/ml IGF1. To replicate the conditions used for imaging, we treated cells with either PPP or IGF1 for 30 min, washed the cells with medium and added fresh medium containing PPP or IGF1. After a further 30 min, neurons were lysed and samples assessed for activation of Akt and Erk1/2. A significant decrease in Akt activation was detected in PMN treated with PPP compared to controls (100% to 70.1 ± 10.7%; Fig 3A and B). However, PPP had no significant effect on Erk1/2 activation (100% to 77.3 ± 9.3%; Fig 3A and B). In agreement with our data, PPP has been previously shown to preferentially decrease Akt activation instead of Erk [21]. When PMNs were treated with IGF1, we found a robust increase in Akt activation (100% to 135 ± 5.7%; Fig 3A and B) without a significant change in Erk1/2 (100% to 102 ± 14.2%; Fig 3A and B). However, it has been shown that activation of Erk1/2 peaks around 10 min after IGF1 stimulation [25,26], and therefore, we may be outside a suitable temporal window to observe Erk1/2 activation in our experimental conditions.

Given the heterogeneity of PMN cultures, we measured pAkt (S473) specifically in neurons using immunofluorescence. As shown in Fig EV2A and B, treatment with 1 μM PPP caused a significant decrease in pAkt mean staining intensity compared to DMSO controls (100% to 66.6 ± 10.2%), thus confirming the results obtained with biochemical approach described above (Fig 3A and B). Surprisingly, when we incubated PMN with 50 ng/ml IGF1, the change in pAkt did not reach significance (100% to 112 ± 8.3%; Fig EV2A and B). This result may be partially explained by the presence of growth factors in our culture medium, which would determine a steady-state activation of Akt in control conditions.

To explore whether the decrease in Akt activation is responsible for the increase in retrograde transport of signalling endosomes, we used two specific commercially available Akt inhibitors (Capivasertib and Ipatasertib). We measured their ability to inhibit Akt in PMN and N2A cells via the phosphorylation of TSC2, a protein which is directly controlled by Akt activation [27]. In both cell types, both Capivasertib and Ipatasertib decreased TSC2 phosphorylation (Fig EV2C–E). To assess retrograde axonal transport, we treated DIV6-7 PMN with either 10 nM Capivasertib or 10 nM Ipatasertib. This lead to a significant increase in signalling endosome transport compared to PMN treated with DMSO (DMSO: 1.38 ± 0.03 μm/s, Capivasertib: 1.70 ± 0.03 μm/s, Ipatasertib: 1.80 ± 0.03 μm/s, Fig 3C and Appendix Fig S2A and B, Movies EV3–EV5). This increase was due to both a decrease in pausing events (DMSO: 24.8 ± 2.22%, Capivasertib: 20.4 ± 1.65%, Ipatasertib: 17 ± 2.14%, Fig 3D and Appendix Fig S2C) and an increase in the instantaneous velocity (Appendix Fig S2D). Overall, these data suggest that the activation state of IGF1R influences the rate of retrograde transport of signalling endosomes and that this effect may be exerted via a modulation of the PI3k/Akt pathway.

**IGF1R knockdown increases the retrograde velocity of signalling endosomes**

To confirm the results obtained by pharmacological inhibition of IGF1R, we next generated a lentiviral shRNA vector targeting this receptor. PMNs were treated with the shRNA viral vector 6 h after

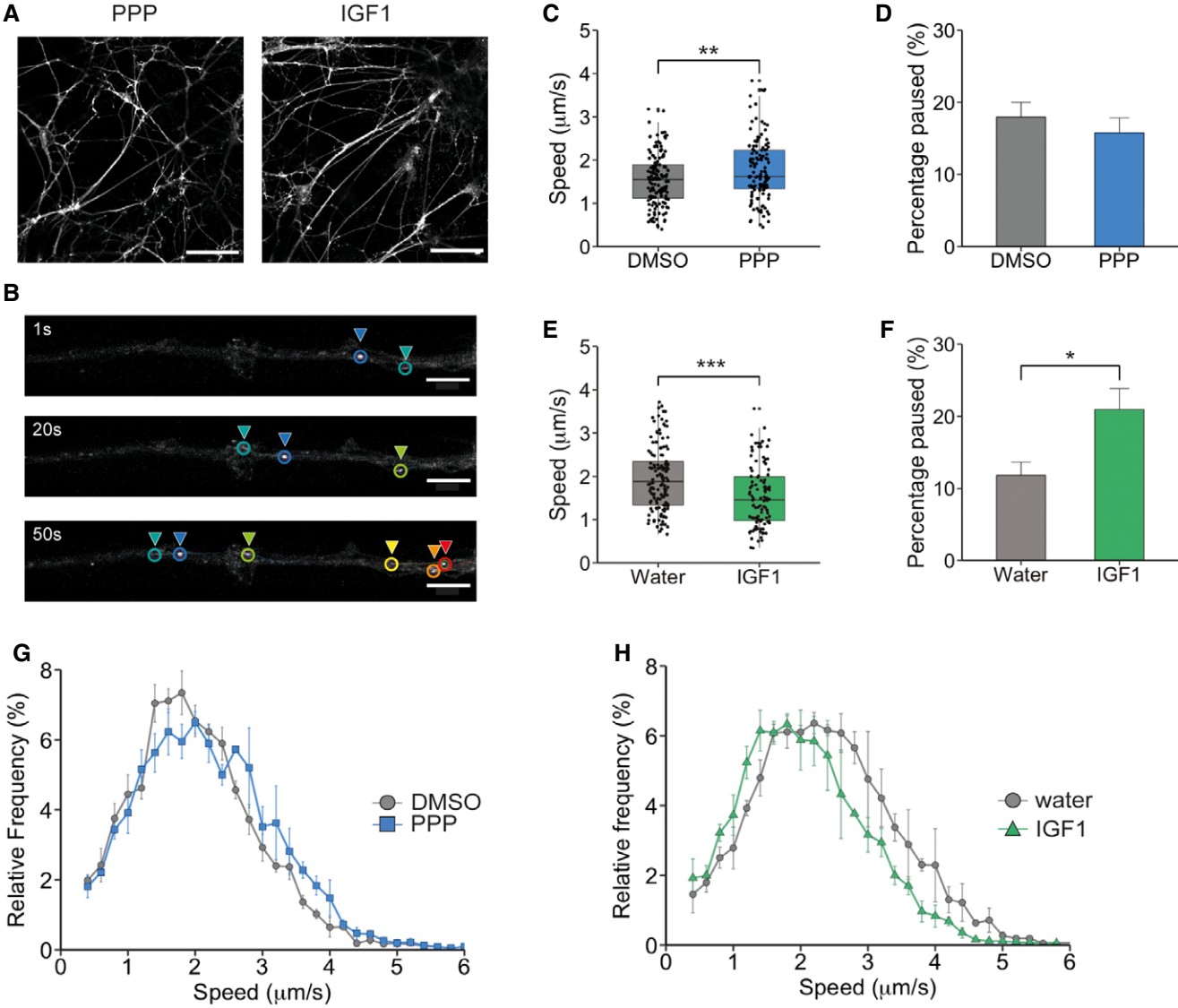

**Figure 2. The IGF1R pathway influences retrograde axonal transport of signalling endosomes.**

PMNs were treated with 1 μM PPP (blue) or 50 ng/ml IGF1 (green) for 30 min before assessing axonal transport using 30 nM Alexa Fluor 555-H$_c$T.

A Example images of DIV6 PMN treated with 1 μM PPP or 50 ng/ml IGF1. PMNs were stained with 30 nM Alexa Fluor 555-H$_c$T. The scale bar is 50 μm.

B Example of H$_c$T-containing organelle tracking in the axon of PMN. Each colour denotes a single endosome tracked over time. The scale bar is 10 μm.

C Graph shows the average velocity of H$_c$T-containing organelles, after PPP treatment compared to controls (DMSO: 138 endosomes, 24 axons, 6,433 movements; PPP: 130 endosomes, 23 axons, 5,222 movements) (**$P = 0.0061$, Student's $t$-test, $N = 4$ independent experiments; boxplot shows median, first and third quartiles. Upper/lower whiskers extend to 1.5 * the interquartile range).

D Per cent of time H$_c$T-containing organelles pausing per axon. There was no difference in pausing between PPP-treated neurons and controls ($P = 0.45$, Student's $t$-test, $N = 4$ independent experiments; data shown are mean ± SEM).

E Graph shows the average velocity of H$_c$T-containing organelles after IGF1 treatment compared to control (water: 138 endosomes, 22 axons, 5,119 movements; IGF1: 106 endosomes, 20 axons, 5,202 movements; ***$P = 2.83 \times 10^{-5}$, Student's $t$-test, $N = 3$ independent experiments; boxplot shows median, first and third quartiles. Upper/lower whiskers extend to 1.5 * the interquartile range).

F Per cent of time H$_c$T-containing organelles spent pausing per axon. IGF1 significantly increased the pausing time of H$_c$T-containing organelles (*$P = 0.011$, Student's $t$-test, $N = 3$ independent experiments; data shown are mean ± SEM).

G Speed distribution profile of PPP-treated neurons and controls. PPP caused an increase in instantaneous velocities of H$_c$T-containing organelles ($N = 3$ independent experiments; data shown are mean ± SEM).

H Speed distribution profile of IGF1-treated neurons and controls. IGF1 caused a decrease in instantaneous velocities of H$_c$T-containing organelles compared to controls ($N = 3$ independent experiments; data shown are mean ± SEM).

plating, and pIGF1R levels were assessed at DIV 6. Cells treated with IGF1R shRNA showed a robust decrease in pIGF1R protein levels when compared to the scrambled shRNA control (Fig EV3A and B).

Next, we assessed the effect of IGF1R knockdown on retrograde signalling endosome transport in PMN. Reducing IGF1R levels caused a significant increase in the mean velocity of signalling

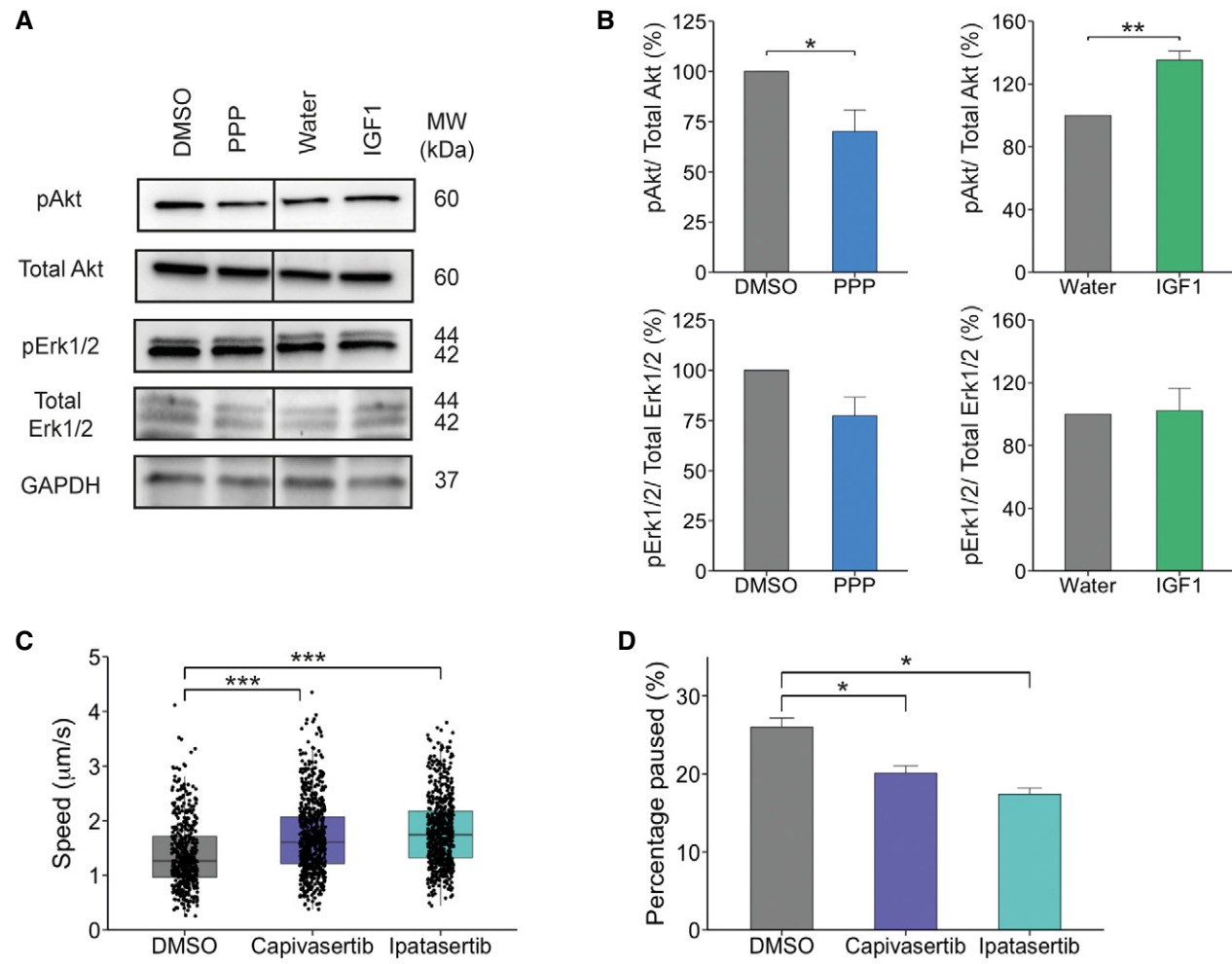

**Figure 3. PPP downregulates Akt signalling.**

A   Western blot showing the levels of pAKT (Ser473), total Akt, pErk1/2 (T202/Y204) and total Erk1/2 after treatment with 1 μM PPP or 50 ng/ml IGF1 for 60 min. Protein was normalized to GAPDH.

B   Western blot quantification. PMN treated with PPP showed a significant decrease in Akt activation (*$P$ = 0.049, $N$ = 5 independent experiments, Student's $t$-test), but no significant difference was seen on Erk1/2 activation ($P$ = 0.07, Student's $t$-test, $N$ = 5 independent experiments). 50 ng/ml IGF1-treated neurons showed a significant increase in Akt activation (**$P$ = 0.008, Student's $t$-test, $N$ = 3 independent experiments). In contrast, no change in Erk1/2 activation ($P$ = 0.88, Student's $t$-test, $N$ = 3 independent experiments) was observed (data shown are mean ± SEM).

C   Graph shows the average velocity of H$_C$T-containing organelles treated with Capivasertib or Ipatasertib (10 nM) for 30 min compared to controls (DMSO: 483 endosomes, 34 axons, 41,897 movements; Capivasertib: 658 endosomes, 43 axons, 49,841 movements; Ipatasertib: 605 endosomes, 32 axons, 41,485 movements) (**$P$ = 2 × 10$^{-16}$, one-way ANOVA, Tukey's post hoc test: DMSO-Capivasertib and DMSO-Ipatasertib ***$P$ < 0.0001; $N$ = 6 independent experiments; boxplot shows median, first and third quartiles. Upper/lower whiskers extend to 1.5 * the interquartile range).

D   Per cent of time H$_C$T-containing organelles pausing per axon. Treatment with Ipatasertib and Capivasertib reduced the amount of time organelles spent pausing (*$P$ = 1.14 × 10$^{-7}$, one-way ANOVA, Tukey's post hoc test: DMSO-Capivasertib $P$ = 9.37 × 10$^{-5}$, DMSO-Ipatasertib $P$ = 2 × 10$^{-7}$; data shown are mean ± SEM, $N$ = 6 independent experiments).

endosomes (1.57 ± 0.07 μm/s to 1.85 ± 0.09 μm/s; Figs 4A and EV3C and D) and a shift towards higher speeds in the instantaneous velocity profile (Fig 4B) compared to the scrambled shRNA control, therefore confirming the effects observed upon pharmacological inhibition (Fig 2C and G). Taken together, these results demonstrate that both short- and long-term modulation of IGF1R activity regulates retrograde axonal transport in PMN and confirm that the effects observed following treatment of PMN with IGF1R-directed small-molecule inhibitors are not due to pharmacological off-target effects of these compounds.

## The role of IGF1R in axonal microtubule dynamics

Given the crucial role of microtubules in axonal transport, we next investigated whether the effects of IGF1R modulation on retrograde transport velocities are due to altered microtubule dynamics. Axonal microtubules are highly polarized, with their plus-end positioned towards the distal end of the axon. Since microtubule polarity is essential for the correct trafficking of organelles [28], alterations in microtubule stability or polarity are likely to impact on the retrograde transport of signalling endosomes. To assess

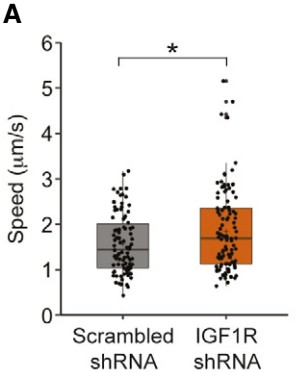
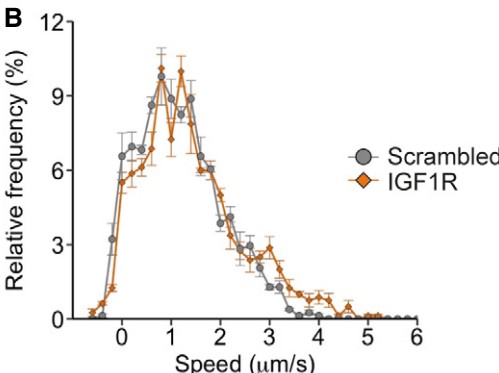

**Figure 4. Knockdown of IGF1R increases retrograde axonal transport of signalling endosomes.**

A PMNs were treated with either an shRNA targeting IGF1R (1 μl) or a scrambled control 6 h after plating. At DIV 6–7, retrograde axonal transport was analysed using 30 nM Alexa Fluor 555-H$_c$T. Knockdown of IGF1R caused a significant increase in the average velocity of signalling endosomes compared to controls (IGF1R shRNA: 108 endosomes, 21 axons, 801 movements; scrambled shRNA: 94 endosomes, 20 axons, 777 movements, *P = 0.011, Student's *t*-test, N = 3 independent experiments; boxplot shows median, first and third quartiles. Upper/lower whiskers extend to 1.5 * the interquartile range).

B Speed distribution profile including pausing events of PMN treated with a shRNA targeting IGF1R or a scrambled control. IGF1R knockdown caused an increase in instantaneous velocities of H$_c$T-containing organelles and decreased the amount of pausing (N = 3 independent experiments; data shown are mean ± SEM).

microtubule dynamics, we magnetofected DIV 5 PMN with a GFP-EB3 construct. End-binding (EB) proteins bind to the plus-end of growing microtubules [29] and consequently enable the analysis of microtubule dynamics in living cells. Following treatment of PMN with 1 μM PPP and 50 ng/ml IGF1, we found that there was no difference in axon polarity in either treatment or control groups (Fig EV4A–D; ~95% of EB3 comets displayed anterograde directionality). To our knowledge, this is the first determination of microtubule polarity in PMN and confirms the highly organized architecture of the microtubule cytoskeleton among different neuronal types.

Intriguingly, we found that modulating the activation of IGF1R significantly altered EB3 comet velocity. Inhibition of IGF1R decreased the velocity of EB3 comets (0.14 ± 0.04 μm/s to 0.09 ± 0.03 μm/s, Movies EV6 and EV7), whilst treatment with IGF1 caused an increase in velocity compared to controls (0.10 ± 0.02 μm/s to 0.14 ± 0.04 μm/s; Fig EV4E). These results suggest that the IGF1R pathway regulates microtubule dynamics in the axon and corroborates previous studies showing that IGF1 is essential for neuronal polarity [13,14] and induces axonal outgrowth [30].

Given the alteration in speed of microtubule plus-end movement, we next sought to understand whether the modulation of IGF1R activity would alter the post-translational modifications of tubulin. Tyrosinated α-tubulin is important for the initiation step of dynein–dynactin processivity [31] and cargo binding to microtubules [32]. However, treatment with PPP or IGF1 did not change the level of tyrosinated or detyrosinated α-tubulin compared to controls (Appendix Fig S3A–C), nor did they cause an overall microtubule loss (Appendix Fig S3D).

Taken together, these results indicate that although IGF1R influences some aspects of microtubule dynamics (e.g. Fig EV4E), it does not lead to changes in microtubule polarity or post-translational modifications of tubulin in the time frame of our experiment (Fig EV4A–D and Appendix Fig S3). It is therefore unlikely that these effects are responsible for the observed changes in signalling endosome velocity.

## Retrograde transport of mitochondria and lysosomes is not affected by IGF1R activity

To investigate the specificity of IGF1R modulation on axonal transport, we next analysed the motility of mitochondria and lysosomes, organelles which, like signalling endosomes, are powered by cytoplasmic dynein along microtubule tracks [8]. We hypothesized that if the IGF1R signalling pathway also influenced the retrograde transport of these organelles, it may directly affect the activity or levels of cytoplasmic dynein. To assess this possibility, PMNs were cultured in microfluidic chambers [33] and treated with either 1 μM PPP or 50 ng/ml IGF1 at DIV 6 prior to assessing mitochondrial motility. We found that under these conditions, the bi-directional transport of mitochondria was not modulated by IGF1R activity (Fig EV5A–D). A large proportion (~50%) of mitochondria were stationary throughout the experiment, with a slight preference towards anterograde movements compared to retrograde (Fig EV5A, B, E and F). Treatment with PPP or IGF1 had no effect on the retrograde transport velocity of mitochondria (Fig EV5C), suggesting that cytoplasmic dynein is not the target of IGF1R activity. In contrast, incubation with IGF1 caused a significant inhibition of the anterograde transport of mitochondria compared to controls (Fig EV5D), suggesting a possible modulation of IGF1 on kinesin-dependent transport.

Similarly, to mitochondria, lysosomes are largely stationary organelles in PMN axons and undergo bi-directional transport, with a preference towards retrograde movements. Kinetic analysis of lysosomal trafficking in PMN axons did not reveal any change in mean velocity between treatment groups and controls (Appendix Fig S4), suggesting that IGF1R activity specifically regulates the axonal transport of signalling endosomes.

## IGF1R inhibition alters key components of dynein-mediated transport

Given IGF1R's selective effect on the retrograde transport of signalling endosomes, we next analysed the effects of IGF1R inhibition

and activation on cytoplasmic dynein. Previous studies have shown that different adaptor proteins modulate transport speeds by recruiting additional dynein complexes to the cargo [34]. We therefore tested whether modulation of IGF1R results in a change in the cellular levels of cytoplasmic dynein and its adaptors. We found no significant change in expression levels of either cytoplasmic dynein or the anterograde motor protein kinesin-1 in lysates from PMN treated with 1 μM PPP or 50 ng/ml IGF1 (Appendix Fig S5). These results, together with the lack of changes in mitochondrial or lysosomal trafficking, suggest that IGF1R does not directly modulate the cellular levels of cytoplasmic dynein and kinesin-1, but rather works via an alternative mechanism.

The Hook family of proteins has been found to play an important role in the retrograde transport of endosomes [35–37]. Therefore, we decided to analyse the expression levels of Hook3 in PMNs following modulation of IGF1R activity, along with two other dynein adaptors: BICD1 and BICD2. Although we saw no change in Hook3 protein expression following either 1 μM PPP or 50 ng/ml IGF1 treatment (Fig 5A and B), PPP treatment did result in a ~37% increase in BICD1 levels (Fig 5A and C), whilst the decrease induced by IGF1 was not significant. BICD1 has been shown to be important for the endosomal sorting of internalized neurotrophin receptors [16] and the retrograde transport of Rab6-positive organelles [38]. This effect is specific, since no changes in the expression levels of the closely related BICD2 adaptor were detected (Fig 5A and D). These findings suggest that IGF1R activity controls the axonal transport rates of signalling endosomes by regulating the cellular levels of the dynein adaptor BICD1.

## The increase in BICD1 expression is due to an upregulation of its protein synthesis

To explore the mechanism underlying the increase in BICD1 levels following IGF1R inhibition, we treated PMN with cycloheximide (CHX) (50 μg/ml) or MG-132 (10 μM) 30 min prior to treatment with 1 μM PPP. CHX is a potent inhibitor of protein translation [39], whereas MG-132 inhibits the 26S proteasome [40], thus blocking protein degradation. Compared to PMN treated with PPP alone, CHX caused a significant decrease in BICD1 expression levels, whilst MG-132 had no significant effect (Fig 6A). These findings suggest that the change in BICD1 protein levels observed after IGF1R inhibition is caused by an increase in its translation. However, if BICD1 is responsible for the observed changes in signalling endosome velocity, BICD1 should be locally synthesized in the axon to enable the short kinetics (30–90 min) of these IGF1R-mediated effects. To test this hypothesis, we performed a puromycin-proximity ligation assay (PLA), a technique which allows the quantification of newly synthesized proteins at specific sites within a cell [41]. For this experiment, we cultured PMN in a 3-compartment microfluidic chamber and then quantified the amount of puromycin-PLA dots per field of view after PPP treatment (Fig 6B). In line with previous results, we saw a significant increase in newly synthesized BICD1 in the axon following PPP treatment when compared to vehicle controls (Fig 6C and D, Appendix Fig S6). This increase was abolished when cells were co-treated with 40 μM anisomycin, a potent protein synthesis inhibitor (Fig 6C and D, Appendix Fig S6). These results corroborate our previous findings that BICD1 expression is upregulated after IGF1R inhibition and demonstrates that this occurs in the axon

within a time frame compatible with the IGF1R-mediated regulation of the retrograde axonal transport of signalling endosomes.

## Inhibition of IGF1R increases *in vivo* signalling endosome motility in a mouse model of ALS

Since IGF1R inhibition was found to enhance retrograde axonal transport, we hypothesized that it may be able to correct the transport deficits that have been previously detected in the SOD1[G93A] mouse model of ALS [3]. Therefore, to explore the effects of IGF1R inhibition on retrograde axonal transport *in vivo*, early symptomatic SOD1[G93A] mice at 72–73 days of age [3,12] and wild-type littermates were anaesthetized and Alexa Fluor 555-H$_c$T (13 μg) and BDNF (50 ng) were injected into the tibialis anterior and gastrocnemius muscles. At the same time, PPP (5 mg/kg) or vehicle control (1% methylcellulose) was administered via intraperitoneal injection. Mice were allowed to recover for 4 h and re-anaesthetized, and retrograde axonal transport was measured in the exposed sciatic nerve [3] (Movie EV8). As PPP has been shown to be virtually non-toxic in animal models [22,42], it was therefore a valuable test compound for this *in vivo* approach. Retrograde axonal transport was analysed 4 h after PPP treatment in the sciatic nerve of live, anaesthetized mice [43]. Firstly, we confirmed that at 73 days of age, the SOD1[G93A] mouse exhibits a deficit in signalling endosome transport compared to wild-type littermate controls (2.14 ± 0.13 μm/s versus 2.59 ± 0.09 μm/s; Fig 7A and B, Appendix Fig S7A) [3]. Interestingly, IGF1R inhibition not only increased retrograde signalling endosome velocity in wild-type mice, from 2.59 ± 0.09 μm/s to 2.90 ± 0.08 μm/s, but also improved axonal transport in the SOD1[G93A] mouse, from 2.14 ± 0.13 μm/s to 2.38 ± 0.17 μm/s (Fig 7A and B, Appendix Fig S7A). As observed *in vitro* (Fig 2C, D and G), this effect is due to an increased instantaneous velocity (Appendix Fig S7B and C) and not due to a change in signalling endosome pausing (Fig 7B). To confirm this change in velocity was due to IGF1R inhibition, sciatic nerves were removed after imaging and analysed for downstream IGF1R signalling (Appendix Fig S7D). Intriguingly, whilst we did not observe a significant change in Akt activation in the sciatic nerve after PPP treatment (Fig 7C), we did detect a significant decrease in Erk1/2 activation in both genotypes (Fig 7D).

Altogether, our findings clearly demonstrate the importance of IGF1R activity in the control of retrograde transport of signalling endosomes both *in vitro* and *in vivo*. Furthermore, since inhibition of IGF1R is able to restore the deficits in axon transport in motor axons of SOD1[G93A] mice, our results also identify IGF1R as a new node for therapeutic intervention in ALS.

## Discussion

Regulation of neuronal trafficking is essential for neuronal maintenance and survival. In this study, we have identified IGF1R as a modulator of signalling endosome transport, providing further insights into the mechanisms which regulate this process. To date, multiple protein kinases have been found to alter axonal transport [44]. In our study, inhibition of IGF1R leads to a significant increase in signalling endosome speed. We observed this effect both in cultured PMN and *in vivo*, in the sciatic nerve of live, anaesthetized

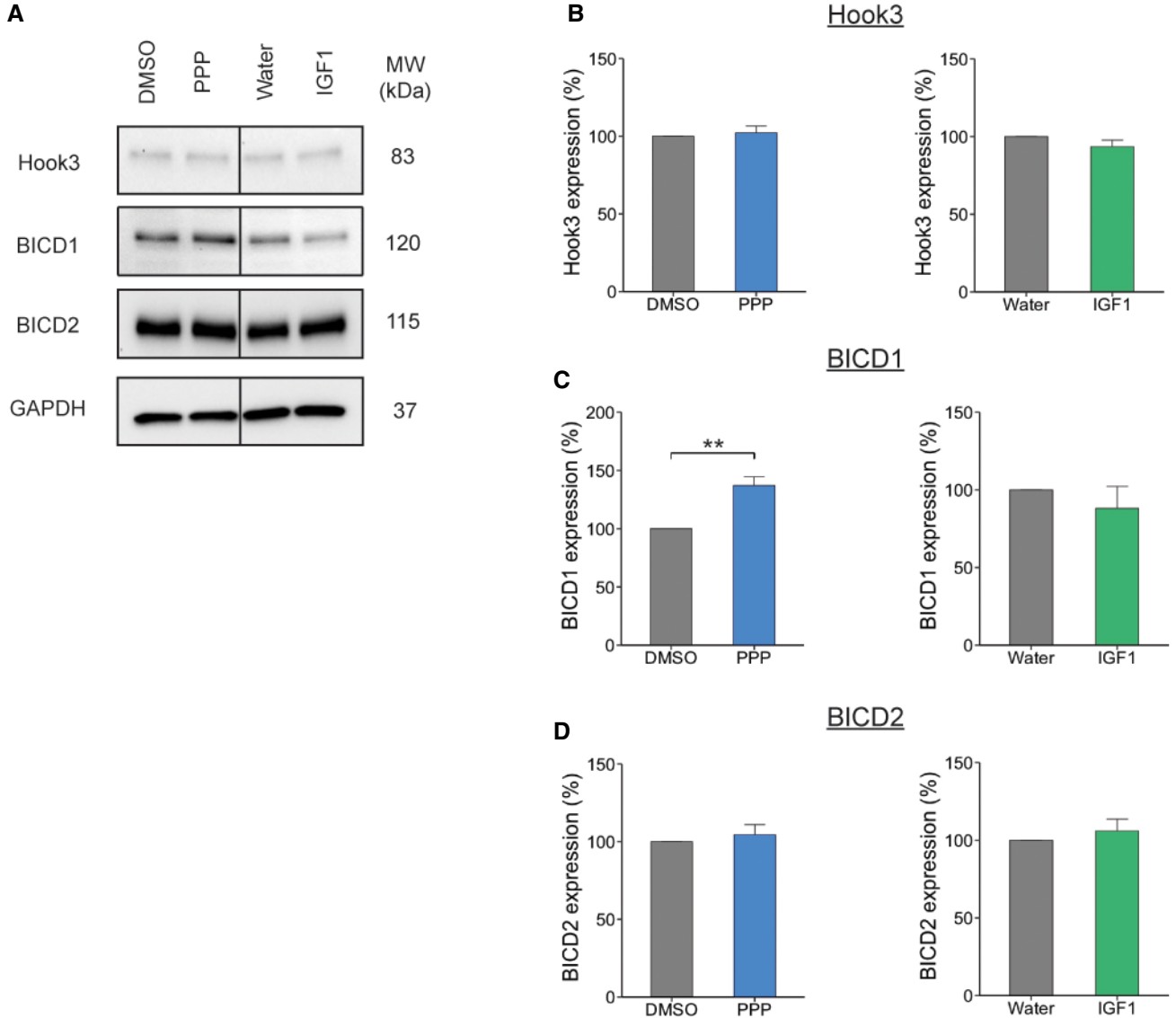

**Figure 5. IGF1R inhibition alters the expression level of BICD1.**

A Western blot of PMN treated with 1 μM PPP or 50 ng/ml IGF1 for 60 min. Cells were lysed and the extracts assessed for expression levels of dynein adaptor proteins, all proteins were assessed from the same blot.

B Hook3 expression did not change after treatment with PPP or IGF1 compared to controls (DMSO, 100%; PPP, 102.16 ± 4.3%, P = 0.67, Student's *t*-test, N = 3 independent experiments; water, 100%; IGF1, 93.48 ± 4.2%, P = 0.26, N = 3 independent experiments; data shown are mean ± SEM).

C However, BICD1 expression was significantly increased in PMN treated with PPP compared to control (DMSO, 100%; PPP, 137.1 ± 7.3%, **P = 0.002, Student's *t*-test, N = 7 independent experiments). In contrast, IGF1R stimulation by IGF1 had no effect on BICD1 expression (water, 100%; IGF1, 88.1 ± 14.0%, P = 0.49, Student's *t*-test, N = 3 independent experiments; data shown are mean ± SEM).

D BICD2 expression did not change after treatment with PPP or IGF1 compared to controls (DMSO, 100%; PPP, 104.44 ± 6.5%, P = 0.51, N = 3 independent experiments: water, 100%; IGF1, 106 ± 7.5%, P = 0.51, Student's *t*-test, N = 3 independent experiments; data shown are mean ± SEM).

Source data are available online for this figure.

mice. This effect was mainly due to an increase in instantaneous velocity and not by a decrease in pausing. Previously, kinases such as CDK5 have been shown to influence motor protein binding, with inhibition of the kinase leading to the phosphorylation of kinesin light chains via GSK3 activation, thus effecting their ability to bind cargo [45]. On the other hand, some kinases have been shown to alter transport via phosphorylation of motor binding proteins. For

example, Colin *et al* found that activation of the PI3K/Akt pathway caused the phosphorylation of huntingtin, leading to increased recruitment of kinesin-1 to the dynactin complex on BDNF-containing vesicles [46].

To establish which pathways were involved in the changes in signalling endosome velocity identified in this study, we investigated the downstream components of the IGF1R signalling cascade.

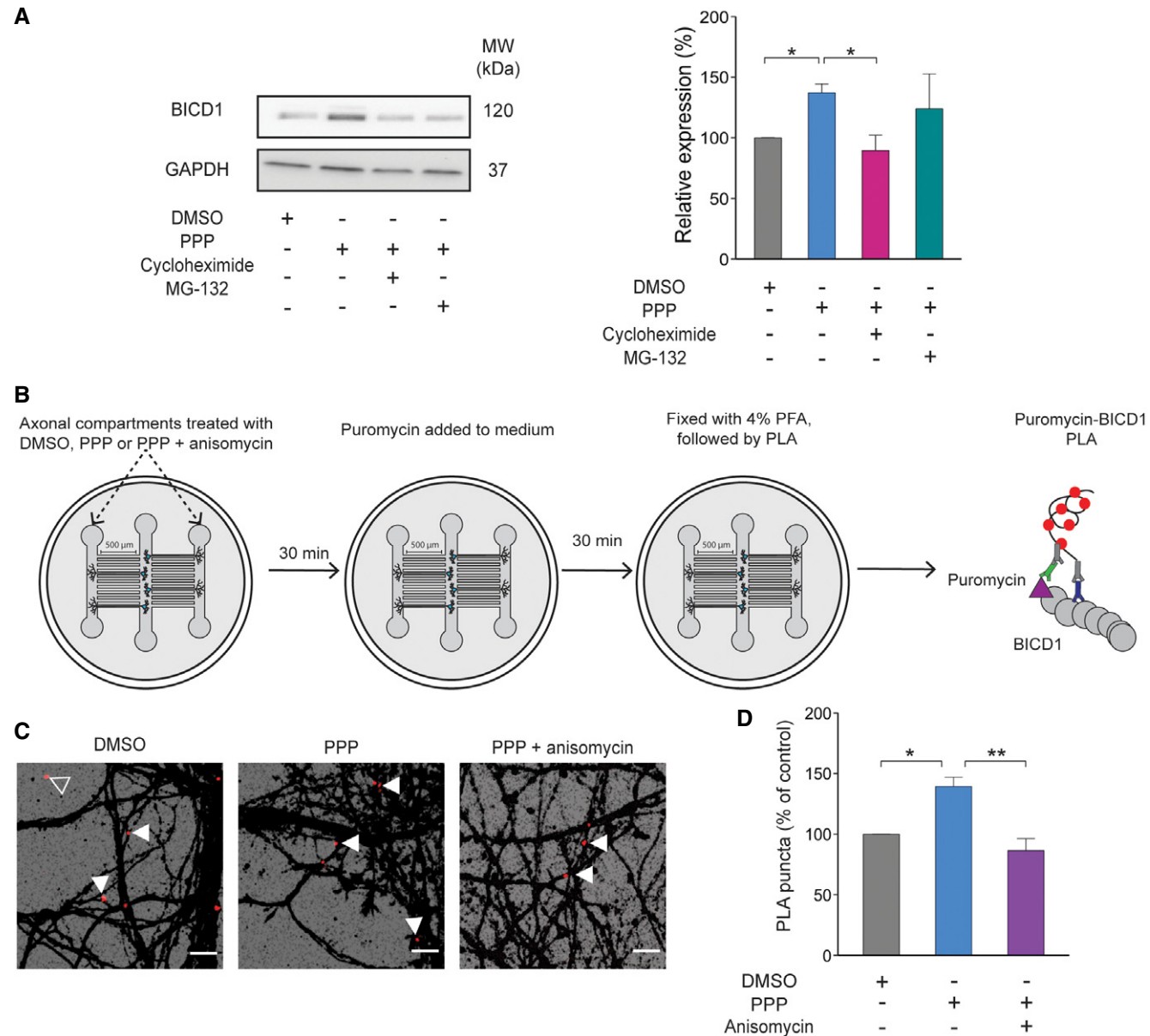

Figure 6. **BICD1 expression in axons is upregulated by *de novo* synthesis.**

A   Western blot of PMN pre-treatment with cycloheximide (50 μg/ml) or MG-132 (10 μM) for 30 min before incubation with 1 μM PPP for 60 min. Cycloheximide causes a significant decrease in BICD1 levels compared to PPP-treated cells. (DMSO, grey: 100%, $N$ = 7 independent experiments; PPP, blue: 137.1 ± 7.3%, $N$ = 7 independent experiments; CHX, pink: 89.5 ± 12.7%, $N$ = 5 independent experiments; MG-132, teal: 124 ± 28.7, $N$ = 3 independent experiments; $P$ = 0.009, one-way ANOVA; Tukey's post hoc test: DMSO-PPP *$P$ = 0.04, PPP-PPP+CHX *$P$ = 0.01; data shown are mean ± SEM).

B   Schematic of puromycin-PLA. Cells were treated with DMSO, PPP or PPP + anisomycin for 30 min in the axonal compartment. After this, puromycin was added to the medium for an additional 30 min. Cells were then fixed, and the PLA protocol was performed (see Materials and Methods). Puromycin—purple triangle; BICD1—grey circles; anti-puromycin antibody—green; anti-BICD1 antibody—blue; secondary antibodies—black.

C   Puromycin-PLA signal in PMN grown in microfluidic devices. Images are magnifications from the white boxes in Appendix Fig S6 taken from the axonal compartment. The scale bars are 10 μm. White arrows point to puromycin-PLA puncta within the β3-tubulin mask; the empty arrow shows a PLA puncta not associated with neurons.

D   Quantification of puromycin-PLA signal. PPP causes an increase in PLA puncta in the axon of PMN, which is abolished by treatment with 40 μM anisomycin (DMSO, grey: 100%; PPP, blue: 139.3 ± 7.7%; PPP + anisomycin, purple: 86.3 ± 10.0; $P$ = 0.005, one-way ANOVA, Tukey's post hoc test; DMSO versus PPP, *$P$ = 0.02; PPP versus PPP + anisomycin, **$P$ = 0.005, $N$ = 3 independent experiments, 27-57 images; data shown are mean ± SEM).

Intriguingly, we found evidence for the involvement of both the Akt and Erk1/2 pathways. *In vitro*, we observed a significant decrease in Akt activation after IGF1R inhibition. Furthermore, when we used specific Akt inhibitors, we also saw an increase in signalling endosome retrograde transport. Akt has been previously shown to alter axonal transport by promoting the binding of kinesin-1 to BDNF-

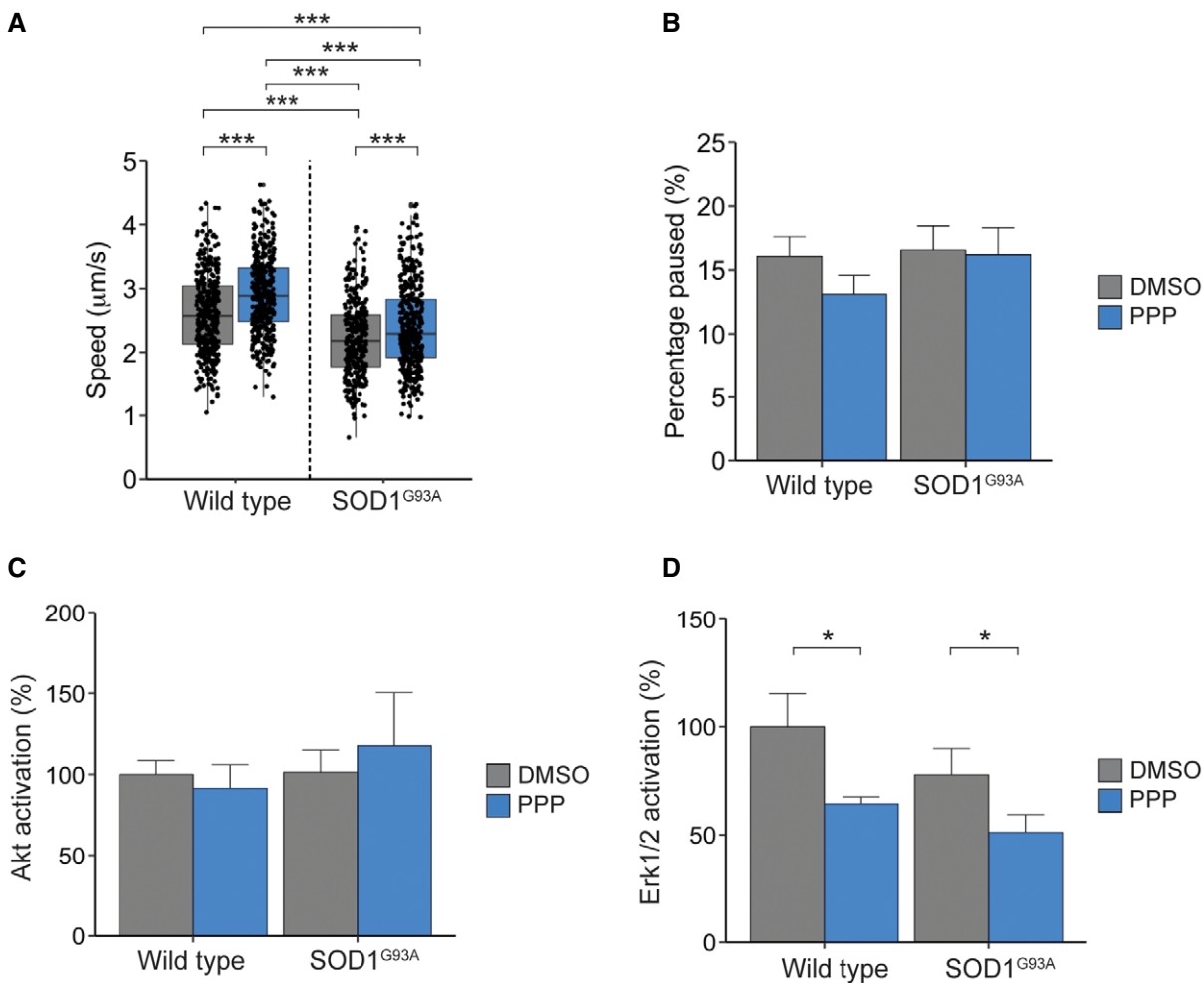

**Figure 7. IGF1R inhibition increases the rate of retrograde axonal transport in an ALS mouse model.**

A   The graph shows the average velocity of $H_cT$-containing organelles *in vivo* in D72/73 SOD1$^{G93A}$ mice and wild-type littermate controls. PPP (blue) treatment increased the transport rate of signalling endosomes in wild-type and SOD1$^{G93A}$ mice compared to animals treated with vehicle control (grey) (WT control: cargo 371, movements 15,272, $N$ = 4 independent experiments; WT + PPP: cargo 388, movements 14,512, $N$ = 4 independent experiments; SOD1$^{G93A}$ control: cargo 322, movements 15,425, $N$ = 4 independent experiments; SOD1$^{G93A}$ + PPP: cargo 404, movements 17,643, $N$ = 4 independent experiments; $P$ = 2.24 × 10$^{-12}$ (treatment), $P$ = 2.22 × 10$^{-16}$ (genotype), two-way ANOVA, Tukey's post hoc test: all conditions ***$P$ < 0.005; boxplot shows median, first and third quartiles. Upper/lower whiskers extend to 1.5 * the interquartile range).

B   PPP treatment does not alter the pausing of signalling endosomes (WT: 16.1 ± 1.5%, WT + PPP: 13.1 ± 1.5%; SOD1$^{G93A}$: 16.6 ± 1.9%, SOD1$^{G93A}$ + PPP: 16.2 ± 2.1%; $P$ = 0.36 (treatment), $P$ = 0.32 (genotype), $N$ = 4 independent experiments for each condition; data shown are mean ± SEM.

C   Treatment with PPP had no effect on Akt levels in the sciatic nerve (WT: Control-100 ± 8.6, PPP-91.5 ± 14.6%; SOD1: Control-101.4 ± 13.5%, PPP-117.6 ± 32.9%, $P$ = 0.81 (treatment), $P$ = 0.53 (genotype), two-way ANOVA, $N$ = 5 independent experiments; data shown are mean ± SEM, Appendix Fig S7D).

D   Erk1/2 activation was downregulated after treatment with PPP in the sciatic nerve of wild-type and SOD1$^{G93A}$ mice (WT: control-100 ± 15.4%, PPP-64.4 ± 3.2%; SOD1: Control-78.0 ± 11.9%, PPP-51.2 ± 8.2%, *$P$ = 0.013 (treatment), $P$ = 0.13 (genotype), two-way ANOVA, $N$ = 5 independent experiments; data shown are mean ± SEM, Appendix Fig S7D). Protein was normalized to GAPDH protein levels.

containing vesicles [46]. Therefore, IGF1R inhibition could lead to the reduction of kinesin-1 on signalling endosomes, resulting in faster retrograde transport. Furthermore, GSK3β inhibition, a downstream target of IGF1R, leads to a decrease in anterograde transport of mitochondria [47]. Since IGF1R activation has been shown to inhibit GSK3β [48], this result could explain the decrease in the anterograde axonal transport of mitochondria we detected in this study following IGF1 treatment.

However, when assessing downstream signalling components in the sciatic nerve after IGF1R inhibition, we found that Erk1/2

activation was significantly impaired. The difference between the effects of IGF1R inhibition *in vitro* and *in vivo* may be due to the different time points at which the activation of the components of the IGF1R signalling cascade was analysed, as PPP has been shown to alter Erk1/2 phosphorylation after prolonged treatment (24 h) [42].

Interestingly, Erk1/2 has been previously implicated in the regulation of transport. Mitchell *et al* showed that activation of Erk1/2 by BDNF caused the phosphorylation of dynein intermediate chain, promoting its recruitment to signalling endosomes [10]. If this was the case, IGF1R inhibition should lead to a slow-down in signalling

endosome trafficking, with reduced Erk1/2 activity causing a decrease in the recruitment of dynein intermediate chain to signalling endosomes. However, we experimentally observed an increase in signalling endosome velocity. This result may, at least in part, be explained by the difference in the neuronal population examined in these studies, with Mitchell *et al* using cortical and hippocampal neurons compared to motor neurons in the present study. Nevertheless, these findings highlight the complex nature of axonal transport and the cell-specific pathways underlying its regulation.

Despite evidence that some kinases regulate transport via directly influencing dynein [10,11], our results suggest that this is not the case for IGF1R. In this study, we observed a specific alteration in signalling endosome transport following IGF1R inhibition, whilst the retrograde trafficking of mitochondria or lysosomes was unaffected. It should be noted, however, that lysotracker can accumulate in other acidic compartments, such as late endosomes and autophagosomes. However, given that we see no change in velocity after PPP treatment, we believe this further highlights the specificity of IGF1R-dependent regulation of signalling endosomes. Additionally, we observed no change in the cellular levels of cytoplasmic dynein after IGF1R inhibition. Furthermore, when investigating the potential role of the microtubule cytoskeleton on the retrograde transport of signalling endosomes, we found that IGF1R does not alter microtubule polarity or post-translational modifications, but does change the growth rate of microtubules. Interestingly, a recent study has highlighted the importance of stable microtubules for efficient retrograde transport [49].

Post-translational modifications of tubulin have also been found to play a key role in regulating intracellular transport. However, we found no significant difference in tyrosination or detyrosinated tubulin levels after PPP treatment of PMN. In spite of the change in growth rate, these findings suggest that the microtubule cytoskeleton is not responsible for the changes in velocity. This possibility is supported by a previous study in which treatment of neurons with a low dose of nocodazole, a potent microtubule depolymerization agent, altered the levels of tyrosinated α-tubulin, but did not lead to changes in the cellular trafficking of BDNF-containing vesicles [50].

A potential regulatory mechanism not investigated in this study is how microtubule-associated proteins (MAPs) can influence transport. With regard to dynein-mediated trafficking, tau has been found to negatively regulate the transport of BDNF-containing vesicles, with the removal of tau from the axon increasing BNDF transport [51]. Interestingly, this mechanism is controlled by the phosphorylation of tau. Given that IGF1R-null mice have been found to have hyperphosphorylated tau [52], it is possible that acute modulation of IGF1R could alter the presence of tau on microtubules, thus influencing transport. Findings from recent work implicate MAPs further and demonstrate that the MAP she1 can directly influence the ATPase activity of dynein and reduce the stepping frequency of the motor [53]. The precise role of MAPs in modulating retrograde axonal transport after IGF1R modulation should be investigated in the future.

Dynein's adaptor proteins play a critical role in determining cargo-specificity and the kinetics of the dynein–dynactin cargo interactions. We discovered that IGF1R inhibition significantly influenced the cellular levels of BICD1, suggesting that changes in BICD1 levels may be the mechanism that underlies the changes in velocity we observed. This is in line with a recent study highlighting the importance of BICD1 in regulating axonal cargo transport [54]. The change in BICD1 levels was driven by newly synthesized axonal BICD1 and occurred within a compatible time frame. Previous results support this hypothesis, with Hafner *et al* finding that 10-min BDNF treatment can significantly alter the local transcriptome of pre- and post-synaptic compartments [55]. Furthermore, Villarin *et al* demonstrated that neurons are capable of locally translating dynein adaptors to help meet the demands of retrograde transport, with nerve growth factor stimulating the upregulation of Lis1 in DRG neurons [56].

How IGF1R inhibition leads to the increase in BICD1 synthesis is currently unknown. The activation of IGF1R and its downstream targets, Akt and ERK1/2, has been shown to play key roles in neuronal protein synthesis through the mTOR pathway [57,58], but further studies will be required to fully understand this process. Interestingly, IGF1R inhibition may also stabilize the interaction between BICD1 and dynein as GSK3β activity has been shown to influence this interaction [59].

In this study, we found that treatment with PPP significantly increased the retrograde transport of signalling endosomes in the SOD1$^{G93A}$ mouse model of ALS *in vivo*, restoring transport rates to near wild-type levels (2.59 ± 0.09 μm/s versus 2.38 ± 0.17 μm/s, WT and SOD1$^{G93A}$+PPP, respectively). Unfortunately, we were unable to test in this study the effects of chronic PPP treatment as long-term administration of methylcellulose used as a PPP carrier causes severe side-effects, including hepato- and splenomegaly [12]. However, given that deficits in axonal transport have been implicated as a key pathological mechanism in the development of ALS [3,12], these data point to IGF1R as a new target for therapeutic intervention. Interestingly, IGF1 has long been thought of as a key neuroprotective factor, with studies showing that treatment with IGF1 improves the lifespan in models of neurodegeneration [60]. Furthermore, abnormal energy homeostasis has been widely implicated in ALS, with hypermetabolism, energy deficits and alterations of lipid metabolism being present both in models of ALS and patients [61,62]. However, IGF1R inhibition has been shown to be extend lifespan and delay the onset of age-related disease [63]. Additionally, two studies have found that the inhibition of IGF1R can lead to significant improvements in lifespan and alleviate symptoms in mouse models of spinal muscular atrophy and Alzheimer's disease [64,65]. Given these findings, furthering our understanding of how IGF1R can have such polar effects will be key to correctly modulate this pathway in neurodegenerative diseases. It will also be important to understand the long-term effects of IGF1R inhibition in disease progression and whether a correction of transport deficits can delay the onset of symptoms.

In conclusion, we have identified IGF1R as a key regulator of retrograde axonal transport in motor neurons both *in vitro* and *in vivo*. This effect was specific to signalling endosomes, as IGF1R inhibition did not affect the retrograde transport rates of mitochondria or lysosomes. PPP was found to increase the rate of BICD1 protein synthesis in motor axons, suggesting that increased BICD1 levels may be responsible for the increase in retrograde velocities detected upon IGF1R inhibition. This pathway was also shown to improve the axonal transport of signalling endosomes in an established ALS mouse model. Taken together, our results point to IGF1R as an important target for therapeutic intervention in neurodegenerative disorders such as ALS.

# Materials and Methods

### Animals and tissue collection

All experiments were carried out following the guidelines of the UCL Institute of Neurology Genetic Manipulation and Ethics Committees and in accordance with the European Community Council Directive of 24 November 1986 (86/609/EEC). Animal experiments were carried out under license from the UK Home Office in accordance with the Animals (Scientific Procedures) Act 1986 and were approved by the UCL Institute of Neurology Ethical Review Committee. Female transgenic mice heterozygous for mutant human SOD1 gene (G93A) on a C56BL/6-SJL mixed background (B6SJLTg [SOD1*G93A]1Gur/J) and wild-type littermates were used for these experiments. Mice were genotyped for the human SOD1[G93A] transgene from ear or tail genomic DNA. Female mice at 72–73 days of age were randomly allocated into vehicle (1% methylcellulose; Sigma-Aldrich M7140i, St Louis, MO)- or PPP (Tocris, Bristol, UK)-treated groups for the experiment shown in Fig 7.

### Plasmids and reagents

Chemicals were from Sigma-Aldrich unless otherwise stated. Compound E4 (2-{[2-({4-chloro-2-methoxy-5-[(1-propyl-4-piperidinyl)oxy]phenyl}amino)-1H-pyrrolo[2,3-d]pyrimidin-4-yl]amino}-6-fluorobenzamide; GSK1713088A) was provided by GSK (UK). PPP and IGF1 were purchased from Tocris and PromoKine (Heidelberg, Germany), respectively. The Akt inhibitors Capivasertib and Ipatasertib were purchased from MedChemExpress (Sweden). The chosen IGF1R shRNA plasmid was purchased from OmicsLink™ (GeneCopoeia, Rockville, MD) and was inserted in a psi-LVRU6GP plasmid with an eGFP reporter gene.

The GFP-EB3 plasmid was kindly donated by Dr M. Way (The Francis Crick Institute, London, UK). Antibodies used for immunofluorescence (IF) and Western blotting (WB) were as follows: Erk1/2 (9102, 1:1,000), phospho-Erk1/2 (T202/Y204; 9101, 1:1,000), Akt (9272, 1:1,000) and phospho-Akt (S473; D9E XP 4060, WB - 1:1,000, IF - 1:400) (all from Cell Signaling); phospho-IGF1R (Tyr 1161/1165/1166; ABE332, 1:500), GAPDH (AB2302, 1:5,000), puromycin (MabE343, 1:2,000), dynein intermediate chain (Mab1618, clone 74.1, 1:500), kinesin heavy chain/KIF5 (Mab1614, 1:100), tyrosinated tubulin (Mab1864, clone YL1/2, 1:500) and detyrosinated tubulin (AB3201, 1:500) (all from EMD Millipore); BICD1 (HPA041309, 1:400) and BICD2 (HPA023013, 1:500) from Atlas Antibodies; and β3-tubulin (Synaptic Systems, 302305, 1:500), Hook3 (Santa Cruz, C-10, 1:500) and GFP (Cancer Research UK, 4E12/8, 1:1,000).

### Motor neuron culture

PMNs were isolated from wild-type E12.5-13.5 mouse embryos. Briefly, embryos were euthanized, spinal cord removed and ventral regions isolated. PMNs were dissociated by incubation with trypsin, followed by mechanical dissociation in combination with DNase treatment. Cells were then centrifuged through a bovine serum albumin (BSA) cushion and resuspended in motor neuron media (Neurobasal; Thermo Fisher, Waltham, MA), 2% v/v B27 supplement (Thermo Fisher), 2% heat-inactivated horse serum (HRS), 1X

GlutaMAX (Thermo Fisher), 24.8 μM β-mercaptoethanol, 10 ng/ml rat ciliary neurotrophic factor (CNTF; R&D Systems, Minneapolis, MN), 0.1 ng/ml rat glial cell line-derived neurotrophic factor (GDNF; R&D systems), 1 ng/ml human brain-derived neurotrophic factor (BDNF; PeproTech, UK) and 1% penicillin/streptomycin. PMNs were immediately plated on poly-D, L-ornithine/laminin-coated plates or microfluidic devices and cultured for 6–7 days at 37°C in a 5% $CO_2$ incubator.

### Production of lentiviral shRNA particles and PMN transduction

IGF1R shRNA and scrambled control viral particles were made by co-transfecting shRNA, packaging and envelope plasmids into Lenti-X 293 T cells (ClonTech, Mountain View, CA) with Lipofectamine 3000 (Thermo Fisher). Medium was collected 48 h after transfection and then again 24 h later. Medium containing lentiviral particles was concentrated using a Lenti-X concentrator (Takara, Japan) and resuspended in Opti-MEM (Thermo Fisher). shRNA viral particles were stored at −80°C until needed. PMNs were transduced 6 h after plating with viral particles being added directly to the medium. After 16 h, the medium was replaced and cells were assessed at DIV 6–7.

### Neuronal imaging

*In vitro* live axonal transport assays were performed on DIV 6–7 PMN grown in MatTek dishes (Ashland, MA) [66]. Briefly, cells were incubated with 30 nM Alexa Fluor 555-$H_cT$ or Alexa Fluor 647-$H_cT$ for 30 min at 37°C. Cells were washed, and then, new motor neuron media containing 20 mM HEPES-NaOH (pH 7.4) were added. After 15 min, retrograde transport was assessed at 37°C using an inverted Zeiss LSM 780 microscope equipped with a Zeiss 40×, 1.3 NA DIC Plan-Apochromat oil-immersion objective. Images were taken at 2 Hz over a period of 2–4 min.

Mitochondria and lysosome transport were assessed in PMN grown in microfluidic chambers for 6 d before the addition of 20 nM tetramethylrhodamine, methyl ester (TMRM) or 50 nM LysoTracker Green DND-26™ (both from Thermo Fisher) to the axonal compartment for 30 min at 37°C. Cells were washed, and motor neuron medium containing 20 mM HEPES-NaOH (pH 7.4) was added to both microfluidic compartments. After 15 min, transport of mitochondria and lysosomes was imaged as described above with 1 Hz image acquisition over a period of 4 min.

For EB3 imaging experiments, neurons were plated on 25-mm coverslips and cultured until 5 DIV. Before magnetofection, the medium was replaced with transfection medium (2% B27, 1X GlutaMAX, 0.04% β-mercaptoethanol, 10 ng/ml GDNF, 10 ng/ml BDNF, 10 ng/ml CNTF in Neurobasal). The old medium was kept and mixed at a 50:50 ratio with fresh medium. Cells were left to acclimatize for 1 h before the addition of the transfection reagents [MEM, 1.5 μg EB3-GFP plasmid DNA and 1.75 μl NeuroMag (Oz Bioscience, France), 1:4 ratio to the transfection media]. Cells were then placed on a magnet for 15 min at 37°C in a 5% $CO_2$ incubator. After this, the magnet was removed and cells were kept in the same environment for 30 min. The magnetofection medium was then replaced with pre-warmed 50:50 old/new medium mix. Cells were incubated at 37°C in a 5% $CO_2$ incubator for 5 h before imaging using an Attofluor® cell chamber (Thermo Fisher) using a Zeiss

63×, 1.3 NA DIC Plan-Apochromat oil-immersion objective. An image was taken every second over a period of 3 min.

Lysosome transport was assessed in PMN grown in microfluidic chambers for 6 d before the addition of 50 nM LysoTracker Green DND-26™ (Thermo Fisher) to the axonal compartment for 30 min at 37°C. Cells were washed, and motor neuron medium containing 20 mM HEPES-NaOH (pH 7.4) was added to both microfluidic compartments. After 15 min, transport of lysosomes was imaged at 37°C using an inverted Zeiss LSM 780 microscope equipped with a Zeiss 40×, 1.3 NA DIC Plan-Apochromat oil-immersion objective. Images were taken at 1 Hz image acquisition over a period of 4 min.

### In vivo axonal transport retrograde assay

*In vivo* axonal transport assays were performed on 72- to 73-day-old wild-type (non-transgenic) and SOD1$^{G93A}$ mice, as previously described [43]. Briefly, mice were anaesthetized and Alexa Fluor 555-H$_c$T (13 μg) and BDNF (50 ng) were injected into the tibialis anterior and gastrocnemius muscles. Mice were injected intraperitoneally (i.p) with either 5 mg/kg PPP or an equivalent volume of vehicle control (1% methylcellulose). The mouse was then left to recover for 4 h, re-anaesthetized and then the sciatic nerve exposed before placing the animal on a heated stage in an environmental chamber at 37°C. Axonal transport in the intact sciatic nerve was imaged on an inverted Zeiss LSM 780 microscope equipped with a Zeiss 40×, 1.3 NA DIC Plan-Apochromat oil-immersion objective. Images were taken at 2 Hz over a period of at least 5 min.

### Immunofluorescence

PMNs were fixed in 4% paraformaldehyde (PFA) in PBS for 12 min at room temperature. Coverslips were washed with PBS, permeabilized and blocked for 15 min using a solution of 0.5% BSA, 10% HRS, 0.2% Triton X-100 in PBS. Primary antibodies were diluted in a solution of 0.5% BSA, 10% HRS in PBS and incubated with the cells for 1 h at room temperature. Cells were then washed in PBS and incubated for 1 h at room temperature with the appropriate fluorescently conjugated secondary antibody (1:1,000) and DAPI (1:2,000) diluted in 0.5% BSA, 10% HRS in PBS. Finally, cells were washed with PBS, mounted using DAKO mounting media and imaged using an inverted Zeiss LSM 780 or 510 confocal microscopes using a 63×, 1.4 NA DIC Plan-Apochromat oil-immersion objective.

### Puromycin-proximity ligation assay

Neurons were cultured in tripartite microfluidic devices for 6 days. Puro-PLA was carried out as previously described [41]. Briefly, cells were treated with either DMSO, 1 μM PPP or 1 μM PPP + 40 μM anisomycin for 30 min. After this, cells' 2 μM puromycin was added to the medium for 30 min at 37°C in a 5% CO$_2$ incubator. Cells were then put on ice and washed with ice-cold 0.2 M acetic acid, 0.5 M NaCl, pH 3.2 (acid wash solution), followed by two quick washes with ice-cold PBS. After fixation, cells were washed again and permeabilized as described in the "Immunofluorescence" section. Detection of newly synthesized proteins was carried out by an anti-

puromycin antibody, an anti-BICD1 antibody and Duolink reagents according to the manufacturer's instructions. For these experiments, rabbit PLA$^{plus}$ and mouse PLA$^{minus}$ were used. Anti-β3-tubulin antibodies were used as a cell marker and to identify axons as described in the "Immunofluorescence" section.

### Western blotting

PMN and sciatic nerves were lysates were prepared in RIPA buffer (50 mM Tris–HCl pH 7.5, 150 mM NaCl, 1% NP-40, 0.5% sodium deoxycholate, 0.1% SDS, 1 mM EDTA, 1 mM EGTA) containing Halt™ phosphatase and protease inhibitor cocktail (1:100, Thermo Fisher) and left to incubate on ice for at least 30 min. Lysates were then spun at 14,800× *g* for 20 min at 4°C. Supernatant were collected, added to 4× Laemmli buffer (250 mM Tris–HCl pH 6.8, 8% SDS, 40% glycerol, 0.02% bromophenol blue, 10% β-mercaptoethanol) and denatured at 100°C for 4 min. Protein concentration was determined using the BCA assay (Thermo Fisher). Samples were run on precast 4–15% gradient gels (Bio-Rad) and transferred onto a polyvinylidene difluoride (PVDF) membrane (Bio-Rad, CA, USA). Membranes were blocked in 5% BSA or milk dissolved in Tris-buffered saline containing 0.1% Tween-20 (TBST) for 1 h at room temperature and then incubated with primary antibody diluted in 5% BSA or milk overnight at 4°C. Blots were washed and then incubated with the appropriate horseradish peroxidase-conjugated secondary antibody for 1 h at room temperature. Immunoreactivity was detected using chemiluminescent substrates (EMD Millipore) and the ChemiDoc™ Touch Imaging System equipped with the ImageLab software (Bio-Rad).

### Image analysis

Image analysis was done using Fiji (NIH, MD). For the analysis of signalling endosome dynamics, TrackMate imaging software [67] and R [68] were used. Output data from this analysis consist of link and track data. The link data contain information of each individual movement performed by a cargo, whilst the track data summarize the overall movement of a cargo. Pauses were defined as the organelle moving slower than 0.2 μm/s. For mitochondrial, lysosomal and EB3 motility, kymographs were generated and analysed using KymoAnalyzer imaging software [69]. Lines on the kymographs were drawn manually. Moving cargo was classified as having speeds above 0.1 μm/s. This change in pause classification was due to the overall slower speed of mitochondria and lysosomes. For Puro-PLA, the number of dots within the axon was counted and the average per image was calculated. For pAkt, tyrosinated and detyrosinated α-tubulin intensity, β3-tubulin was used as a mask and the integrated intensity measured in Fiji. For tyrosinated and detyrosinated α-tubulin, integrated intensity was measured relative to β3-tubulin.

### Statistical analysis

Statistical analysis was performed in R. Student's *t*-test was used when comparing two groups. A one-/two-way analysis of variance (ANOVA) was used, followed by Tukey's multiple comparison test when multiple groups were present. Statistical significance is noted

as follows: $*P \leq 0.05$, $**P \leq 0.01$, and $***P \leq 0.001$. All statistical tests used and associated p values are indicated in figure legends.

Expanded View for this article is available online.

## Acknowledgements

We thank the personnel of the Denny Brown Laboratories, UCL Queen Square Institute of Neurology, for the maintenance of mouse colonies. We also want to thank Drs J Sleigh, A Tosolini and N Birsa for expert supervision and help. This work was supported by a Leonard Wolfson PhD studentship (ADF), a GSK-BBSRC Industrial CASE studentship (KLG), the European Union's Horizon 2020 Research and Innovation programme under grant agreement 739572 [GS], a Wellcome Trust Senior Investigator Award (107116/Z/15/Z) [GS] and a UK Dementia Research Institute Foundation award [GS]. LG is the Graham Watts Senior Research Fellow supported by Brain Research UK (BRUK).

## Author contributions

GS and LG conceived the work; ADF, KLG and GS designed the experiments; ADF, ERR and KLG performed the experiments; ADF analysed the data; ADF, ERR, LG and GS contributed to the writing of the paper; and all authors have approved submission of this work. The funders had no role in study design, data collection and analysis, decision to publish, or manuscript preparation.

## Conflict of interest

The authors declare that they have no conflict of interest.

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
