## [Review Process File · EMBO Reports]

IGF1R regulates retrograde axonal transport of signalling endosomes in motor neurons

Alexander D. Fellows, Elena R. Rhymes, Katherine L. Gibbs, Linda Greensmith and Giampietro Schiavo

Review timeline:

Submission date:	22 August 2019
Editorial Decision:	2 September 2019
Revision received:	15 November 2019
Editorial Decision:	19 December 2019
Revision received:	23 December 2019
Accepted:	15 January 2020

Transaction Report: This manuscript was transferred to *EMBO reports* following peer review at *The EMBO Journal*

1st Editorial Decision

2 September 2019

Thank you for the transfer of your manuscript and the associated referee reports from The EMBO Journal to EMBO reports.

As we had discussed earlier, we would like to invite you to revise your manuscript for EMBO reports. Please address all technical concerns from the referees and provide all missing control experiments and revise the manuscript along the lines proposed in our earlier communication. These revisions include a clearer definition of pauses and the kymograph analysis, a clarification regarding the different timepoints used and a discussion on how IGF1R might regulate BICD1 protein levels. Please also provide the full list of factors found in the retrograde transport screen, provide further evidence whether AKT inhibition has an effect on axonal transport and whether it mirrors IGF1R inhibition, and provide further data on IGFR1 distribution in primary motor neurons.

Revised manuscripts should be submitted within three months of a request for revision; they will otherwise be treated as new submissions. Please contact us if a 3-months time frame is not sufficient for the revisions so that we can discuss the revisions further.

2) individual production quality figure files as .eps, .tif, .jpg (one file per figure).

Please download our Figure Preparation Guidelines (figure preparation pdf) from our Author Guidelines pages

<https://www.embopress.org/page/journal/14693178/authorguide> for more info on how to prepare

your figures.

4) a complete author checklist, which you can download from our author guidelines (<<https://www.embopress.org/page/journal/14693178/authorguide>>). Please insert information in the checklist that is also reflected in the manuscript. The completed author checklist will also be part of the RPF.

5) Please note that all corresponding authors are required to supply an ORCID ID for their name upon submission of a revised manuscript (<<https://orcid.org/>>). Please find instructions on how to link your ORCID ID to your account in our manuscript tracking system in our Author guidelines (<<https://www.embopress.org/page/journal/14693178/authorguide#authorshipguidelines>>)

6) We replaced Supplementary Information with Expanded View (EV) Figures and Tables that are collapsible/expandable online. A maximum of 5 EV Figures can be typeset. EV Figures should be cited as 'Figure EV1, Figure EV2' etc... in the text and their respective legends should be included in the main text after the legends of regular figures.

7) We would also encourage you to include the source data for figure panels that show essential data. Numerical data should be provided as individual .xls or .csv files (including a tab describing the data). For blots or microscopy, uncropped images should be submitted (using a zip archive if multiple images need to be supplied for one panel). Additional information on source data and instruction on how to label the files are available <<https://www.embopress.org/page/journal/14693178/authorguide#sourcedata>>.

8) Our journal encourages inclusion of *data citations in the reference list* to directly cite datasets that were re-used and obtained from public databases. Data citations in the article text are distinct from normal bibliographical citations and should directly link to the database records from which the data can be accessed. In the main text, data citations are formatted as follows: "Data ref: Smith et al, 2001" or "Data ref: NCBI Sequence Read Archive PRJNA342805, 2017". In the Reference list, data citations must be labeled with "[DATASET]". A data reference must provide the database name, accession number/identifiers and a resolvable link to the landing page from which the data can be accessed at the end of the reference. Further instructions are available at <<https://www.embopress.org/page/journal/14693178/authorguide#referencesformat>>.

9) Regarding data quantification:

- Please ensure to specify the name of the statistical test used to generate error bars and P values, the number (n) of independent experiments underlying each data point (not replicate measures of one sample), and the test used to calculate p-values in each figure legend. Discussion of statistical methodology can be reported in the materials and methods section, but figure legends should contain a basic description of n, P and the test applied.

IMPORTANT: Please note that error bars and statistical comparisons may only be applied to data obtained from at least three independent biological replicates. If the data rely on a smaller number of replicates, scatter blots showing individual data points are recommended.

- Graphs must include a description of the bars and the error bars (s.d., s.e.m.).
- Please also include scale bars in all microscopy images.

10) As part of the EMBO publication's Transparent Editorial Process, EMBO reports publishes online a Review Process File to accompany accepted manuscripts. This File will be published in conjunction with your paper and will include the referee reports, your point-by-point response and all pertinent correspondence relating to the manuscript.

I look forward to seeing a revised version of your manuscript when it is ready. Please let me know if you have questions or comments regarding the revision.

1st Revision - authors' response

15 November 2019

Response to Reviewers' and Editor's comments

The Editor and the Reviewers asked us to clarify various aspects of the experimental design. Furthermore, they required to highlight possible mechanisms at the basis of PPP effects.

To address these important points, we have implemented the following changes to the main text of the manuscript:

Material and Methods

The description of the methodology used to quantify of the movement of axonal organelles was deemed to be incomplete. To this end, we clarified our definition of pauses for different organelles such as mitochondria and lysosomes vs signalling endosomes (see page 25 of the revised manuscript). Additionally, on page 25 we have added information on how kymograph traces were manually selected and how the software has been supervised during quantification of the kinetic data.

Results and Figure legends

The manuscript was judged to be not completely clear and lacking important details related to previously published experiments.

We have rewritten the first sentence of the results to improve readability. Furthermore, the text and figures have been scrutinised for inconsistencies regarding the timing used in different experiments.

However, rather than reproducing a vast amount of previously published data, we have extensively referred to a recently published manuscript (Gibbs *et al.* 2018 *Cell Death Dis* **257**, 26-33), which contains a complete description of the small molecule screen shown in Fig. 1, including the list of small molecules tested in this assay, the dose-dependence of the effect and related controls.

As suggested, we have shifted the data shown in Figure 5 to Expanded view Fig 5. However, we would like to stress that this experiment is crucial and should be kept in the manuscript. Indeed, the finding that the IGFR pathway, which is modulated by PPP, does not directly affect the activity of the dynein motor complex is a new finding emerging from the experiments presented in our work, and not previous knowledge in the field or an expected conclusion, as implicitly suggested by the reviewer.

Discussion: We have added further explanations on how IGF1R may regulate the local expression of BICD1, and the possible links of this process to the AKT/mTOR pathway.

Furthermore, we have clearly described in the text on page 18 the likely shortcomings of long-term treatment of SOD1^{G93A} mice with PPP, which are based on the outcome of our previous tests using an established inhibitor of p38 MAPK alpha in the same animal model of ALS (Gibbs *et al.* 2018 *Cell Death Dis* **257**, 26-33). Briefly, the preclinical trial had to be stopped due to cumulative side

effects (e.g. hepatomegaly and splenomegaly) of the drug and/or the vehicle.

Regarding the analysis of lysosomal transport, we have clarified in the discussion that there is very little / no overlap between axonal signalling endosomes containing HcT and lysotracker-positive organelles in primary spinal cord motor neurons.

Last but not least, we have further refined our discussion on the possible mechanism of regulation of axonal transport by IGF1 and suggested possible avenues for future experiments to test our hypotheses.

In addition to extensively modifying the text and figures to improve clarity, as requested *we have provided additional experiments to address specific gaps found in the original manuscript*. In particular:

1. In the revised Fig 3, panels C and D, we now demonstrate that inhibition of axonal AKT signalling modulates axonal transport, thus confirming our proposed mechanism of action downstream to IGF1R inhibition.
2. Representative examples of neurons which have been imaged and quantified, have now been added to Fig. 2, along with an example of the Trackmate image. Representative movies have also been added throughout the manuscript to further document our results and make our conclusions clearer.
3. New data on the distribution of IGF1R in primary wild type spinal cord motor neurons have now been provided in Expanded Figure 1.
4. We have moved some of the less informative figure panels displaying specific axonal transport parameters (e.g. distribution of velocities per axon and per experiment) to the Supplemental material, as suggested (e.g. Supplemental Fig. 1).
5. We agree with the Reviewer that the effect of PPP on EB3 dynamics is interesting and should be further investigated in more detail. However, we also believe that addressing how modulators of EB3 recruitment to the positive end of microtubules affect the retrograde transport of signalling endosomes is not crucial for the conclusions reached in this manuscript and therefore is outside the scope of this work.
6. We apologise with the Reviewers for giving the impression that we omitted essential controls. To this end, the manuscript has been carefully checked and further clarifications have been added, when appropriate.

Original Referees' Reports

Referee #1

In this article the author's demonstrated novel mechanism by which IGF1R regulate specifically axonal transport of signaling endosomes, but not other organelles such as mitochondria and lysosomes. This changes in endosomal trafficking is likely to be due to increase in axonal local synthesis of the dynein adaptor BICD1 in MN's and not via alteration in Dynein expression. Lastly, the authors suggest that inhibition of IGF1R signal cascade can be a possible basis for ALS treatment via rescuing axonal transport process of ALS model.

The manuscript is very interesting, novel, well written and highly contribute to the neuro-cell biology field. It describes a new mechanism that regulate axonal transport via local synthesis.

We thank this reviewer for appreciating our work and supportive comments.

Still there are few concerns to be clarified before publication:

1. How IGF1R regulates BICD1 local synthesis? Is it via AKT/mTOR pathway? The authors need to demonstrate using compartmental chambers, pharmacological and manipulations of AKT/mTOR in axons similar to what was done in figs 3, 6, and 7; or discuss it in the discussion section.

We agree that the mechanism by which IGF1R signalling regulates the local synthesis of BICD1 is very interesting. As suggested, we have tested the hypothesis that AKT activity regulates axonal transport. We found that, similarly to IGF1R inhibitors, treatment of motor neurons with AKT

inhibitors also increases axonal transport of signalling endosomes (new Fig. 3C,D). However, this mechanism seems to be not directly related to the mTOR pathway, since preliminary experiments performed in the laboratory using mTOR inhibitors suggest that mTOR activation has the opposite effects on this transport route (data not shown). As such, this pathway will be analysed in depth in future work.

2. The authors need to publish (if they didn't do it already), a detailed list of all the factors/signaling pathways that were found in the retrograde transport screen (The yellow box factors of fig 1A).

As mentioned above in the letter to the Editor, rather than reproducing a large amount of previously published data, we have extensively referred to a recently published manuscript (Gibbs *et al.* 2018 *Cell Death Dis* **257**, 26-33), which contains a complete description of the small molecule screen shown in Fig. 1, including the list of small molecules tested in this assay, their dose-dependence and related controls.

3. Does inhibition of axonal AKT signaling, alter axonal transport, similar to IGF1R inhibition?

We thank the reviewer for this great suggestion. Treatment of motor neurons with AKT inhibitors also increases axonal transport of signalling endosomes (new Fig. 3C,D), a result which fully supports our conclusions.

4. As axonal transport can be affected at diff parts of the axons, Does inhibition of IGF1R, change axon growth and length?

We did not detect significant changes in axonal length or branching or other aspects of motor neuron morphology upon treatment with IGF1R inhibitors in the timeframe of our experiments (data not shown).

5. Fig7a needs to analyze BICD1 levels in axons following axonal manipulations.

Whilst this suggestion is potentially interesting, we are not sure how these additional experiments will contribute to further proving the link between IGF1R inhibition and the regulation of axonal transport by modulation of axonal dynein adaptors.

6. Indeed it seems that upon inhibition of IGF1R there is more retrograde endosomal transport. But its relevance treating ALS is very weak, and the authors should demonstrate rescue in motor neurons survival and degeneration, especially as IGF1-R was shown to act positively on neuron survival, or discuss it more in the discussion section.

In light of the reviewer's comment, we changed the text to remove any overstatement regarding the use IGF1R inhibition as effective treatment for ALS. Please also see our reply to the Editor's comment pasted above.

Referee #2

The manuscript by Fellows et al. describes novel mechanism of how retrograde axonal trafficking of signaling endosomes is regulated via the IGF1R signaling. The authors demonstrate that IGF1R inhibition specifically increases the motility of signaling endosomes but not lysosomes or mitochondria. The proposed molecular mechanism involves activation of Akt and the local translation of BICD1 which in turn associates with dynein motor and facilitates its processivity. Then the authors move to the mouse ALS model and demonstrate relevance of this pathway in vivo. The manuscript is interesting and timely. It is well written.

We thank the reviewer for appreciating our work and for the supportive comments.

However, I have several technical and experimental concerns which should be addressed in the revised version of the paper.

Major points:

1. The manuscript contains sufficient number of independent experiments and appropriately performed statistics but generally lacks representative examples. This goes through almost all figures. In Figure 1, examples of cells labeled with the HcT and the α -p75NTR upon treatment with active components identified from a small molecule kinase inhibitor screen as well as example of negative control should be included. Images of representative axonal fragments used for the analysis and examples of derived tracks should be added to the panel in Figure 1B. Same applies for the

other figures where Trackmate was used. Why when using Trackmate pauses were defined as the organelle moving slower than 0.2 $\mu\text{m/s}$ whereas in kymograph analysis everything above 0.1 $\mu\text{m/s}$ was considered as moving particle?

We thank the reviewer for these comments. We added representative examples of the experimental results in the several figures (e.g. Fig. 2 A, B, Expanded view, and Supplemental data). Furthermore, we extensively referred to previous publications (e.g. Gibbs *et al.* 2018 *Cell Death Dis* **257**, 26-33), which showed essential controls regarding the screening assay used to identify regulators of axonal transport. Furthermore, we modified the method section to clarify the parameters used for speed quantification and pause analysis.

2. How does IGF1R distribution in primary motor neurons looks like? Is it enriched in the axonal growth cone or present all over axonal membrane?

These data have been added to Expanded view Fig. 1.

3. In Figure 2 and Figure 4 speed of axonal signalling endosomes is displayed as average of all data, then as distribution of velocities per axon and then per experiment. I think only the first panel should be kept in the main figures whereas two other graphs can be moved to the Expanded view. Then there will be enough space to include example images of measured axons and traces.

We have modified the text and figures accordingly, moving non essential panels to Expanded view and supplemental data.

4. The conclusions of this manuscript are strongly relying on kinetic data. It would be helpful if the key experiments are supported by representative Movies.

We thank this reviewer for pointing this out. In light of this comment, we have added eight representative videos displaying examples of axonal transport in vitro and in vivo in different experimental conditions.

5. Figure 5 and Supplemental Figure 4: Kymographs do not look very convincing. Examples of corresponding axons should be added. It seems that signal-to-noise ratio is not that great. I'm wondering how well KymoAnalyzer can detect the lanes? For instance, Figure 5B, water treatment, detected displacements look very different from what one can see on the original kymograph. Same applies for Figure S4A, PPP treatment or in B, water treatment. Why there are white lines on the kymographs? Performing live imaging with higher frame rate (at least 2 fps instead of 1) will improve the quality of kymographs and make analysis more reliable and easier.

We thank this reviewer for pointing this out and for their suggestions. We routinely perform acquisitions at higher frame rates; reassuringly, however, this does not change the results of our quantitative image analysis (data not shown). However, as requested, we have added representative videos to the revised manuscript to better document our imaging conditions. Furthermore, this kymograph has been moved to the expanded view section (Fig. EV5).

6. LysoTracker is not fully specific for lysosomes but also stains other acidic compartments (i.e. late endosomes). Considering the importance of the finding that BICD1 only influences trafficking of signaling endosomes but not lysosomes and mitochondria, this issue should be addressed with addition experiments. For instance, by overexpressing another lysosomal marker (LAMP1) or labeling functional lysosomes with MagicRed.

We completely agree with the reviewers that LysoTracker is not fully specific for lysosomes but stain a variety of acidic compartments. We have now clarified in the discussion that there is very little / no overlap between axonal signalling endosomes containing HcT and LysoTracker-positive organelles in primary spinal cord motor neurons, making the results shown in Supplemental Fig. 4 even more salient.

7. Is there change in BICD1 expression in the SOD1^{G93A} ALS mouse model? Demonstrating this would help further stringent the link between in vitro and in vivo experiments.

We thank the reviewer of this comment. We fully agree that addressing the level of expression of BICD1 in the SOD1^{G93A} mouse model during disease progression would be interesting. Unfortunately, assessing BICD1 levels in sciatic nerves of these mice has been proven difficult, and did not provide reliable results in the timeframe of this revision.

Minor point: The first sentence of the results is heavy and difficult to read. It introduces one of the central experimental tools used in this paper. It would be helpful if the authors rephrase it or better include a schematic demonstrating the HcT and the a-p75NTR assays, even when these approaches were published previously.

We tried to improve readability of the text throughout the manuscript. However, we opted to not include a schematic of the assay since it was previously published.

Referee #3

Fellows and colleagues investigate the role of IGF1R during signaling endosome trafficking in motor neurons. The identification of the kinase inhibitor (E4) as a new modulator of retrograde axonal transport lead authors to hypothesize that inhibition of IGF1R modulates endosomes retrograde trafficking. To assess that the increase in retrograde velocity of signaling endosomes was in effect mediated by the activation of IGF1R, they inhibit the receptor activation by using PPP (IGF1R inhibitor) and they confirm a positive effect on endosomes velocity in PMN cultures while retrograde velocity is reduced by IGF-1 treatment. These data were confirmed by silencing the endogenous IGF1R. The authors show a subsequent down regulation of Akt activity, but not of Erk1/2. In order to establish that the increase in transport velocity, was not due to altered microtubule (MT) dynamics, they overexpress EB3 and analyze the dynamics in living cells. The treatment of motor neurons with PPP did not change the axon polarity but triggered an increase of EB3 comet velocity, suggesting an effect of IGF1R pathway on MT dynamics. Moreover, authors conclude that the treatment with PPP did not alter the trafficking dynamics of mitochondria and lysosomes, suggesting that inhibition of the receptor does not directly affect dynein activity. In support, they found that IGF1R pathway may modulate the levels of an endosomes-specific dynein adaptor (BICD1) thus triggering an increase of BICD1 synthesis when IGF1R is inhibited and promoting the retrograde transport of signalling endosomes. Finally, Fellows and colleagues report that PPP treatment of SOD1G93A mouse model of ALS led to an increase in retrograde signalling endosomes velocity in the exposed sciatic nerves of wild type mice and to an improvement of axonal transport in the ALS mouse model, reporting also a decrease in ERK1/2 activation in both conditions.

The question is of importance. This is an interesting study that uncovers a potential role of IGF1 and IGF1R pathway in the reinforcement of long-range retrograde axonal transport of signalling endosomes, due to the importance of neurotrophins during axonal growth, maintenance and nerve repair. To make their points, the authors have performed simple experiments. However, the content of experiments are limited and authors do not provide the downstream pathway and mechanism by which BICD1 is induced: effect on the promoter, more efficient translation? Stability? In addition, most of the experiments lack appropriate controls.

We are sorry that this reviewer found our experiments limited. However, our work did not aim to investigate the molecular mechanisms responsible for the regulation of BICD1 expression by IGF1R (e.g. direct or indirect effect on promoter or mRNA stability). However, we did check whether this regulation involved an effect on BICD1 protein stability. In Fig. 6A, we have shown that proteasome activity was not involved in the regulation of BICD1 levels in motor neurons treated with PPP.

On this occasion, we would like to reassure the reviewers that we have performed extensive controls to verify our results and support our conclusions.

Another important concern is the use of different timings for each experiment in vitro and also in vivo (30 min in figure 1, 45 min in figure 2, 60 min in figure 3, several days in figure 4, 45 min in figure 5, 60 min in figure 6, 30 min or 60 in figure 7, material and methods says that imaging was performed 15 min after treatment) and the lack of a clear temporal dynamic of IGF1R inhibition on retrograde transport. Therefore, it is difficult to conclude how IGF-1 leads to BICD1 up-regulation.

We apologise with this reviewer about the apparent variability in the timing used for the different experiments. These changes have been carefully considered and adapted to the different protocols used to assess axonal transport *in vitro* and *in vivo* in order to maximise the signal to noise ratio and the reproducibility of the results. We have carefully revised the text to eliminate inconsistencies and explain our choices.

Figure 1 : Why did authors use an antibody against p75 and not TrkB to investigate the trafficking of pro-survival signaling endosomes?

We have previously explored both types of antibodies in the analysis of the axonal transport of signalling endosomes (e.g. Deinhardt *et al.* 2006. *Neuron* **52**, 293-305; Deinhardt *et al.* 2007. *Traffic* **8**, 1736-1749) and obtained largely overlapping results. On this basis, we chose an antibody against the extracellular domain of p75^{NTR} generated in house as one of the two transport probes for our *in vitro* assay.

Authors indicate that E4 is a IGF-1R inhibitor and show that E4 reduces its phosphorylation. What is the effect of IGF-1? Does it increase phosphorylation of the receptor and is the effect reduced by E4 in PMN?

IGF-1 is a known activator of IGF1R and therefore does increase the phosphorylation of the receptor, as documented widely in the literature. E4 has been previously found to prevent IGF1R phosphorylation as shown in Fig. 1C, and therefore should eliminate any effect of IGF1. We unfortunately did not test experimentally this condition.

Figure 2: Panels A,B,C,,E,F,G of Figure 2: The authors showed the same data using 3 different graphs. It is sufficient to use the average of instantaneous velocities for all the n (experiments). Also, IGF1 treatment shows an increase of pausing time, while there is no difference in pausing time with PPP. How do authors explain the change in pausing time when they add IGF1? As for figure 1, authors should show that IGF-1-induced reduction in retrograde trafficking (and pausing) should be blocked by PPP and E4.

As requested, we have modified Fig. 2, shifting some of the panels to Expanded view and supplemental data. We also agree that treatment of IGF1 and PPP/E4 concurrently would be a good experiment to do in the future.

Figure 3: The authors should perform a time course of Akt and ERK activation and not just the 60 minutes time point. As presented here, the experiment is not sufficient to conclude that there is no implication of ERK signaling cascade (especially given the results of figure 8). Moreover, they should use specific inhibitors of the ERK versus Akt pathway to reinforce their findings.

As commented above, treatment of motor neurons with AKT inhibitors also increases axonal transport of signalling endosomes (new Fig. 3C,D).

Figure S2: The effect on EB3 is interesting and should be further investigated in detail. For example, authors stated that IGF1 is essential for neuron polarity and induces axonal outgrowth. Since they did not observe any difference in axon polarity, they may investigate the effect on axon length or growth cone morphology in presence of IGF-1/PPP. Is there a change in MT stability, catastrophe and rescue?

Do modulators of EB3 attachment to the +TIPs modify retrograde velocity of signaling endosomes?

Moreover, the fact that there is no change in tyrosination, underlines the fact that there is not a clear explanation for the decreased velocity of EB3 comets.

We agree with the Reviewer that the effect of PPP on EB3 dynamics is interesting and should be investigated in more detail. However, we also believe that addressing how modulators of EB3 recruitment to the positive end of microtubules affect the retrograde transport of signalling endosomes is not crucial for the conclusions reached in this manuscript and therefore falls outside the scope of this work.

Figure 4: same comment than for figure 2, one graph is sufficient. However, It would be important to test whether IGF-1 still has an effect when the receptor is silenced.

Figure 5: Given the lack of an effect, this should be in supplemental section. Additionally, cytoplasmic dynein is not a DIRECT target of PPP or IGF1, but it is the assembled molecular motor machinery (with adaptors of dynein) that changes between organelles. Authors should therefore investigate the effect of PPP on the transport of other cargoes and not just mitochondria.

As suggested by this Reviewer, we have investigated the effect of IGF1R inhibition on mitochondria and lysosome transport (see Expanded view Fig. 5 and Supplemental Fig. 4). The transport of both organelles is unaffected by PPP treatment, thus confirming our conclusions on the specificity of these effects on signalling endosome dynamics.

Figure 6: more samples should be provided as the effect is rather small. Is the IGF-1 effect blocked by PPP, silencing of the receptor as well as by Akt and ERK inhibitors?

We respectfully disagree with this comment. Although the effect of PPP on BICD1 expression levels is not massive, it is statistically significant, specific for BICD1 and physiologically relevant. On this basis, what would the aim of increasing the number of experiments be? We would like to remind this Reviewer that without a strong scientific justification, we cannot sacrifice additional mice to prepare the primary cultures needed to perform these experiments.

Figure 7A: The WB is not convincing and does not reflect the corresponding graph (% relative expression). Authors should provide more samples (at least 3 per conditions). Experiment also lacks controls (CHX and MG132 alone). Could autophagy be involved as well? From the WB, it seems that MG132 also efficiently blocks the effect of PPP raising doubts about the exact mechanism involved.

Whilst the western blot may provide the impression that MG132 impinges on the effect of PPP on BICD1 levels (Fig. 6A), the quantification provided in Fig. 6B clearly demonstrate that this is not the case. Henceforth, it is unlikely that proteasome activity alters BICD1 dynamics.

Also, authors should investigate whether inhibition of the IGF-1/Akt pathway affects RNA levels of BICD1 using appropriate inhibitors (and not just PPP).

Figure 7C: The PLA experiment is not convincing as we can barely detect the puncta and again the difference seen in the graph does not match the provided images. Negative control of the experiment should be provided.

We respectfully disagree with the reviewer. We have repeated this experiment three times and CHX consistently inhibited the effect of PPP. Furthermore, this effect has been confirmed by a different experimental set-up in Fig. 6D.

Also, what is the effect of anisomycin in somatic compartment???

Based on the effects of PPP on the axonal levels of BICD1 and our focus on axonal retrograde transport, we focussed on the effects of anisomycin in this compartment. Whilst we believe that anisomycin has multiple effects in the soma, it is unlikely that these activities have a significant impact on the axonal levels of BICD1 given the short time frame of the experiment and the topological separation between the two compartments (> 0.5 mm).

If translation occurs in the axonal compartment, authors should provide evidence for the presence of BICD1 RNA in this compartment and for on-site translation.

We agree that exploring the presence of BICD1 mRNA in axons and its on-site translation would be very interesting. We plan to carry out these very demanding experiments both *in vitro* and *in vivo* in the future, since we believe are outside the scope of this manuscript.

Figure 8: Here, I am lost. Authors show the opposite of what they claimed in figure 3 regarding Akt/ERK activation. In conclusion, although authors correct trafficking of endosomes in vivo, they do not provide the exact mechanism nor evidence that this has an effect.

Due to the experimental set-up, these assays were not undertaken within the same timeframe as the *in vitro* experiments, which may explain these divergent results. Exploring this mechanism would be of great interest in the future, especially as specific Akt inhibitors were found to replicate *in vitro* our results with PPP.

2nd Editorial Decision

19 December 2019

Thank you for the submission of your revised manuscript to EMBO reports. It was evaluated again by former referee 1 and 2 and their reports are copied below.

As you will see, both referees are now very positive about the study and request only minor changes to clarify the figures.

Browsing through the manuscript myself, I noticed a few editorial things that we need before we can proceed with the official acceptance of your study.

- You refer to "data not shown" on page 8 of the manuscript. Please note that per our editorial

policies all data described in the manuscript must be shown.

- Please update the references to the numbered format of EMBO reports. The abbreviation 'et al' should be used if more than 10 authors. You can download the respective EndNote file from our Guide to Authors
<https://drive.google.com/file/d/0BxFM9n2IEE5oOHM4d2xEbmpxN2c/view>
- Please provide the Expanded View figures as individual production quality figure files as .eps, .tif, .jpg (one file per figure). It is sufficient to have their legends in the main manuscript file in the section "Expanded View figure legends"
- Appendix: please correct the nomenclature of the figures within the Appendix pdf file to "Appendix Figure Sx" and please provide a title page with a table of contents (incl. page numbers)
- Please provide legends for the movies. To do so create a simple README.txt file with the legend for the movie and then zip the .txt file together with the movie it describes. Then upload the zipped file.
- Routine figure control
- Thank you for providing Source data. I have one question regarding the source data for Figure 5A, BICD1 blot. Can you please indicate which of the bands you used for the figure panel. I cannot locate the bands in the original blot that match the ones in the figure. Moreover, the source data for the BICD2 blot is missing.
- We also noticed that the graphs in Figure 3D and S2C look almost identical. What is the difference between these two experiments? It is not obvious from the legends. Could you please clarify?
- Please shorten the title to 100 characters including spaces and please have a look at the attached Word file with a few more points regarding figure legends that might have been missed earlier (in track changes).
- Finally, EMBO reports papers are accompanied online by A) a short (1-2 sentences) summary of the findings and their significance, B) 2-3 bullet points highlighting key results and C) a synopsis image that is 550x200-400 pixels large (width x height). You can either show a model or key data in the synopsis image. Please note that the size is rather small and that text needs to be readable at the final size. Please send us this information along with the revised manuscript.

REFEREE REPORTS

Referee #1:

I like the manuscript and the authors answers my previous concerns

Referee #2:

The revised version of the manuscript by Fellows et al. has significantly improved. The authors addressed all of my comments and concerns. This work is a great contribution to the field and I really enjoyed reading it. Perhaps one minor point which will probably be addressed by the production team, it would look nicer if panels and labels in the figures are aligned and have better spatial arrangement.

Combined response to the Editor and the Reviewers

We thank the Editor and the Reviewers for their support and insightful comments that through the Reviewing process has greatly improved the overall quality and impact of our findings. We are delighted that our manuscript is now acceptable in principle for publication in EMBO Rep pending minor amendments and clarifications. Please find below our answers to the Editor's and the Reviewers' queries in blue:

Editorial comments

- You refer to "data not shown" on page 8 of the manuscript. Please note that per our editorial policies all data described in the manuscript must be shown.

We apologise for this oversight. After reading again the manuscript we believe that this additional data was not required to support our conclusion. We have therefore removed it.

- Please update the references to the numbered format of EMBO reports. The abbreviation 'et al' should be used if more than 10 authors. You can download the respective EndNote file from our Guide to Authors: <https://drive.google.com/file/d/0BxFM9n2IEE5oOHM4d2xEbmpxN2c/view> We have updated the references following the suggested Endnote style.

- Please provide the Expanded View figures as individual production quality figure files as .eps, .tif, .jpg (one file per figure). It is sufficient to have their legends in the main manuscript file in the section "Expanded View figure legends"

Quality figure files for the Expanded View figures have now been uploaded on the submission webpage. Expanded View figure legends have now been appended to the main manuscript on page 38.

- Appendix: please correct the nomenclature of the figures within the Appendix pdf file to "Appendix Figure Sx" and please provide a title page with a table of contents (incl. page numbers) The nomenclature of the figures within the Appendix have been changed as instructed and a title page with a table of content added on page 1.

- Please provide legends for the movies. To do so create a simple README.txt file with the legend for the movie and then zip the .txt file together with the movie it describes. Then upload the zipped file.

Legends for the movies have been added to the video folder which has been uploaded.

Routine figure control

- Thank you for providing Source data. I have one question regarding the source data for Figure 5A, BICD1 blot. Can you please indicate which of the bands you used for the figure panel. I cannot locate the bands in the original blot that match the ones in the figure. Moreover, the source data for the BICD2 blot is missing.

Thank you for carefully checking the source data and raising this question, which allowed us to spot a previously unnoticed oversight. Whilst the data previously uploaded correspond to the dataset quantified in Fig. 5 and Fig. S5, they were not those shown in panel A. We have now uploaded the original data corresponding to the panel shown in Fig. 5A and Fig.S5 including the correct GAPDH loading controls. We sincerely apologise for any inconvenience that this may have it caused.

- We also noticed that the graphs in Figure 3D and S2C look almost identical. What is the difference between these two experiments? It is not obvious from the legends. Could you please clarify?

These figures are indeed derived from the same dataset. However, in Fig. 3D measurements are plotted per axon, whilst in Fig. S2C they are shown per experiment, as explained in the corresponding Figure legends.

- Please shorten the title to 100 characters including spaces and please have a look at the attached Word file with a few more points regarding figure legends that might have been missed earlier (in track changes).

The title has been shortened and now reads: "IGF1R regulates retrograde axonal transport of signalling endosomes in motor neurons". We have revised the text to take into account all changes and some minor imprecisions, including a small mistake in Fig EV2. A new version of the main

text, Appendix and Fig EV2 have now been uploaded.

- Finally, EMBO reports papers are accompanied online by A) a short (1-2 sentences) summary of the findings and their significance, B) 2-3 bullet points highlighting key results and C) a synopsis image that is 550x200-400 pixels large (width x height). You can either show a model or key data in the synopsis image. Please note that the size is rather small and that text needs to be readable at the final size. Please send us this information along with the revised manuscript.

As requested, we have added a summary completed with bullet points along with the graphically synopsis.

Referee #1:

I like the manuscript and the authors answers my previous concerns

Referee #2:

The revised version of the manuscript by Fellows et al. has significantly improved. The authors addressed all of my comments and concerns. This work is a great contribution to the field and I really enjoyed reading it. Perhaps one minor point which will probably be addressed by the production team, it would look nicer if panels and labels in the figures are aligned and have better spatial arrangement.

We have examined the Figures but are at lost on how to improve them further. We are open to suggestions from the Production Team on how to improve them further.

Corresponding Author Name: Professor Giampietro Schiavo

Manuscript Number: EMBOR-2019-49129-T